

**Synergetic formation of secondary inorganic and organic**
**aerosol: Influence of SO₂ and/or NH₃ in the heterogeneous**
**process**
**Biwu Chu[a, b], Xiao Zhang[c, d], Yongchun Liu[a, b], Hong He[a, b,*], Yele Sun[b,e], Jingkun**
**Jiang[c], Junhua Li[c], Jiming Hao[c]**
[a] State Key Joint Laboratory of Environment Simulation and Pollution Control,
Research Center for Eco-Environmental Sciences, Chinese Academy of Sciences,
Beijing 100085, China
[b] Center for Excellence in Urban Atmospheric Environment, Institute of Urban
Environment, Chinese Academy of Sciences, Xiamen 361021, China
[c] State Key Joint Laboratory of Environment Simulation and Pollution Control, School
of Environment, Tsinghua University, Beijing 100084, China
[d] Nanjing University of Information Science & Technology, Nanjing 210044, China
[e] State Key Laboratory of Atmospheric Boundary Layer Physics and Atmospheric
Chemistry, Institute of Atmospheric Physics, Chinese Academy of Sciences, Beijing
100029, China
*Correspondence to:* Hong He (honghe@rcees.ac.cn)
**Abstract**
The effects of SO₂ and NH₃ on secondary organic aerosol formation have
rarely been investigated together, while the interactive effects between
inorganic and organic species under highly complex pollution conditions
remain uncertain. Here we studied the effects of SO₂ and NH₃ on secondary
aerosol formation in the photooxidation system of toluene/NO$_x$ in the



presence or absence of $Al_2O_3$ seed aerosols in a 2 $m^3$ smog chamber. The
presence of $SO_2$ increased new particle formation and particle growth
significantly, regardless of whether $NH_3$ was present or not. Sulfate,
organic aerosol, nitrate and ammonium were all found to increase linearly
with increasing $SO_2$ concentrations. The increases in these four species
were more obvious under $NH_3$-rich conditions, and the generation of nitrate,
ammonium and organic aerosol increased more significantly than sulfate
with respect to $SO_2$ concentration, while sulfate was the most sensitive
species under $NH_3$-poor conditions. The synergistic effects between $SO_2$
and $NH_3$ in the heterogeneous process contributed greatly to secondary
aerosol formation. Specifically, the generation of $NH_4NO_3$ was found to be
highly dependent on the surface area concentration of suspended particles,
and increased most significantly among the four species with respect to
$SO_2$ concentration under ammonia-rich conditions. Meanwhile, the
absorbed $NH_3$ might provide a liquid surface layer for the absorption and
subsequent reaction of $SO_2$ and organic products, and therefore, enhance
sulfate and secondary organic aerosol (SOA) formation. This effect mainly
occurred in the heterogeneous process and resulted in a significantly higher
growth rate of seed aerosols compared to that without $NH_3$. By applying
positive matrix factorization (PMF) analysis to the AMS data, two factors
were identified for the generated SOA. One factor, assigned to less-
oxidized organic aerosol and some oligomers, increased with increasing
$SO_2$ under $NH_3$-poor conditions, mainly due to the well-known acid
catalytic effect of the acid products on SOA formation in the heterogeneous





process. The other factor, assigned to the highly oxidized organic
component and some nitrogen-containing organics (NOC), increased with
$SO_2$ under a $NH_3$-rich environment, with NOC (organonitrates and NOC
with reduced N) contributing most of the increase.

## Introduction

With the recent rapid economic development and urbanization, the
associated emissions from coal combustion, motor vehicle exhaust and
various industrial emissions have led to highly complex air pollution in
China. Besides the high concentrations of fine particles ($PM_{2.5}$), high
concentrations of $NO_x$, $SO_2$, $NH_3$, and volatile organic compounds (VOCs)
were observed in haze pollution episodes (Liu et al., 2013; Ye et al., 2011;
Zou et al., 2015; Wang et al., 2015). For example, the $SO_2$ concentration in
Jinan, a city in North China, can be as high as 43 ppb in the winter season
(Wang et al., 2015). The high concentrations of precursors resulted in high
concentrations of secondary inorganic and organic species in $PM_{2.5}$ during
haze formation (Yang et al., 2011; Zhao et al., 2013; Dan et al., 2004; Duan
et al., 2005; Wang et al., 2012). There has been no extensive measurement
of $NH_3$ in China despite its extensive emission and increasing trend (Fu et
al., 2015). A few studies reported high concentrations of $NH_3$ (maximum
concentration higher than 100 ppb) in the North China Plain (Meng et al.,
2015; Wen et al., 2015) and many observation data indicated $NH_3$-rich
conditions for secondary aerosol formation, and strong correlations
between peak levels of fine particles and large increases in $NH_3$





concentrations in China (Ye et al., 2011; Liu et al., 2015a). Under this
complex situation, studying the synergistic effects of $SO_2$ and $NH_3$ among
pollutants in secondary aerosol formation is crucial in order to understand
the formation mechanism of heavy haze pollution.
Interactions between inorganic pollutants in secondary aerosol
formation have been investigated extensively. For example, $NO_2$ was found
to increase the oxidation of $SO_2$ in aqueous aerosol suspensions (Tursic and
Grgic, 2001) and on a sandstone surface (Bai et al., 2006). The synergistic
reaction between $SO_2$ and $NO_2$ on mineral oxides was reported (Liu et al.,
2012a) and proposed to explain the rapid formation of sulfate during heavy
haze days (He et al., 2014). The presence of $NH_3$ was also found to enhance
the conversion of $SO_2$ to sulfate in aerosol water and on the surface of
mineral dust or $PM_{2.5}$ (Tursic et al., 2004; Behera and Sharma, 2011; Yang
et al., 2016).
Secondary aerosol formation from coexisting inorganic and organic
pollutants is far more complicated. There have been a few studies that
investigated the effects of $SO_2$ or $NH_3$ on secondary organic aerosol (SOA)
formation. $SO_2$ has been found to enhance SOA yield from isoprene (Edney
et al., 2005; Kleindienst et al., 2006; Lin et al., 2013), $\alpha$-pinene
(Kleindienst et al., 2006; Jaoui et al., 2008), and anthropogenic precursors
(Santiago et al., 2012) due to its acidic aerosol products, which were
thought to either take up organic species (Liggio and Li, 2008, 2006) or
result in the formation of high molecular weight compounds in acid-
catalytic reactions (Liggio et al., 2007; Kleindienst et al., 2006; Santiago



et al., 2012). Besides, sulfate esters were also confirmed as major players
in SOA formation (Schmitt-Kopplin et al., 2010). The effects of $NH_3$ on
SOA formation are relatively poorly understood. In previous studies,
disparate effects of $NH_3$ on secondary aerosol formation were reported. It
was found that the presence of $NH_3$ increased SOA formation in the
reaction of $\alpha$-pinene or cyclohexene with ozone (Na et al., 2007), but had
little effect on SOA mass in isoprene ozonolysis (Na et al., 2007; Lin et al.,
2013) and even decreased SOA production from the reaction of styrene and
ozone (Na et al., 2006). $NH_3$ was reported to react with some organic acids
and contribute to secondary aerosol formation (Na et al., 2007; Lin et al.,
2013), while nucleophilic $NH_3$ might attack and decompose trioxolane and
hydroxyl-substituted esters (Na et al., 2006), and therefore decrease SOA
mass. Updyke et al. (2012) studied brown carbon formation via reactions
of ammonia with SOA from various precursors and emphasized that aging
by $NH_3$ is not a unique mechanism of SOA browning. It was found that the
degree of browning had a positive correlation with the carbonyl products,
which may react with $NH_3$ and generate hemiaminal (Amarnath et al.,
1991), while the form of ammonia ($NH_3$ gas or $NH_4^+$ ion) had little
influence on the browning processes.
The effects of $SO_2$ and $NH_3$ on SOA formation have rarely been
investigated together, while the interactive effects between inorganic and
organic species under highly complex pollution conditions remain
uncertain. This study investigated secondary aerosol formation in the
photooxidation of toluene/$NO_x$ with varied concentrations of $SO_2$ under





NH$_3$-poor and NH$_3$-rich conditions. Some synergetic effects in the
heterogeneous process that contributed to both secondary inorganic and
organic aerosol formation were explored.

## 123 Methods

A series of smog chamber experiments were carried out to simulate
secondary aerosol formation in the photooxidation of VOC/NO$_x$ in the
presence or absence of SO$_2$ and/or NH$_3$. The chamber is a 2 m$^3$ cuboid
reactor constructed with 50 $\mu m$-thick FEP-Teflon film (Toray Industries,
Inc., Japan). The chamber was described in detail in Wu $et$ $al.$ (2007). A
temperature-controlled enclosure (SEWT-Z-120, Escpec, Japan) provides
a constant temperature (30±0.5 ℃), and 40 black lights (GE F40T12/BLB,
peak intensity at 365 nm, General Electric Company, USA) provide
irradiation during the experiments. The hydrocarbon concentration was
measured by a gas chromatograph (GC, Beifen SP-3420, Beifen, China)
equipped with a DB-5 column (30 m×0.53 mm×1.5 mm, Dikma, USA) and
flame ionization detector (FID), while NO$_x$, SO$_2$ and O$_3$ were monitored
by an NO$_x$ analyzer (Model 42C, Thermo Environmental Instruments,
USA), an SO$_2$ analyzer (Model 43I, Thermo Environmental Instruments,
USA) and an O$_3$ analyzer (Model 49C, Thermo Environmental Instruments,
USA), respectively. A scanning mobility particle sizer (SMPS) (TSI 3936,
TSI Incorporated, USA) was used to measure the size distribution of
particulate matter (PM) in the chamber, and also employed to estimate the
volume and mass concentration. The chemical composition of aerosols was



measured by an aerosol chemical speciation monitor (ACSM, Aerodyne
Research Incorporated, USA) or high resolution time of flight aerosol mass
spectrometer (HR-ToF-AMS, Aerodyne Research Incorporated, USA).
ACSM is a simplified version of aerosol mass spectrometry (AMS), with
similar principles and structure. Ng *et al.* (2011) presented a detailed
introduction to this instrument and found that the measurement results
agreed well with the AMS. Wall deposition of particles in the chamber was
similarly corrected using a regression equation to describe the dependence
of deposition rate on the particle size (Takekawa et al., 2003). Detailed
information on this equation was given in our previous studies (Chu et al.,
2012; Chu et al., 2014).

Alumina seed particles were produced on-line via a spray pyrolysis

setup, which has been described in detail elsewhere (Liu et al., 2010).
Liquid alumisol (AlOOH, Lot No. 2205, Kawaken Fine Chemicals Co.,
Ltd., Japan) with an initial concentration of 1.0 wt%, was sprayed to
droplets by an atomizer. After that, the droplets were carried through a
diffusion dryer and a corundum tube embedded in a tubular furnace with
the temperature maintained at 1000 ℃ to generate alumina particles. The
obtained alumina particles were $\gamma$-$Al_2O_3$ as detected by X-ray diffraction
measurements, and spherical-shaped according to electron micrograph
results. Before being introduced into the chamber, the particles were
carried through a neutralizer (TSI 3087, TSI Incorporated, USA). In
addition, toluene was injected into a vaporizer and then carried into the
chamber by purified air, while $NO_x$, $SO_2$ and $NH_3$ were directly injected



into the chamber from standard gas bottles. The concentrations of $NH_3$
were estimated according to the introduced amount of $NH_3$ and the volume
of the reactor.
**Results and discussion**
**Particle formation and growth in different inorganic gas conditions**

The effects of $SO_2$ and $NH_3$ on secondary aerosol formation were

qualitatively studied first in the photooxidation system of toluene/$NO_x$
without the presence of a seed aerosol. Experiments were carried out in the
absence of $SO_2$ and $NH_3$, in the presence of $SO_2$ or $NH_3$, and coexistence
of $SO_2$ and $NH_3$, respectively. Experimental details are listed in Table 1.
The letter codes used for the experiments represent a combination of the
initial letters of the precursors for each experiment. For example,
experiment "ASTN" is an experiment with presence of ammonia gas (A),
sulfur dioxide (S), toluene (T) and nitrogen oxides (N). Two experiments
(ATN1 and ATN2) were carried out under similar conditions to test the
reproducibility of the experiments.

Secondary aerosol formation in these photooxidation experiments was

measured by the SMPS, and the results are displayed in Fig. 1. Assuming
the same aerosol density in these experiments, the presence of either $NH_3$
or $SO_2$ enhanced secondary aerosol formation markedly. Compared to
toluene/$NO_x$ photooxidation, the secondary aerosol volume concentration
rose 1.5 times in the presence of $SO_2$, and was more than tripled in the



presence of the NH$_3$. The volume of secondary aerosol showed an obvious
peak in the toluene/NO$_x$/NH$_3$ system at about 2.3 hours of photooxidation.
With the wall deposition accounted for, the decrease of the volume
concentration after that point was unexpected, but could be reproduced
(Experiment ATN1 and ATN2). Such a decrease was not observed with
coexisting NH$_3$ and SO$_2$, indicating interactions between NH$_3$ and SO$_2$ in
the photooxidation system. The reason for this phenomenon will be
discussed in the following analysis of the chemical composition of the
generated particles.

Table 1  Initial experimental conditions of toluene/NO$_x$ photooxidation in the

presence or absence of SO$_2$ and/or NH$_3$

| Experiment No. | Hydrocarbon *ppm* | NO *ppb* | NO$_x$-NO *ppb* | SO$_2$ *ppb* | NH$_3$[*] *ppb* | RH *%* | T *K* |
|---|---|---|---|---|---|---|---|
| TN | 1.05 | 54 | 49 | 0 | 0 | 50 | 303 |
| STN | 1.05 | 55 | 50 | 137 | 0 | 50 | 303 |
| ATN1 | 1.06 | 47 | 48 | 0 | 264 | 50 | 303 |
| ATN2 | 0.98 | 48 | 54 | 0 | 264 | 50 | 303 |
| ASTN | 1.02 | 49 | 53 | 134 | 264 | 50 | 303 |

[*]The concentrations of NH$_3$ were calculated according to the introduced amount of NH$_3$
and the volume of the reactor.



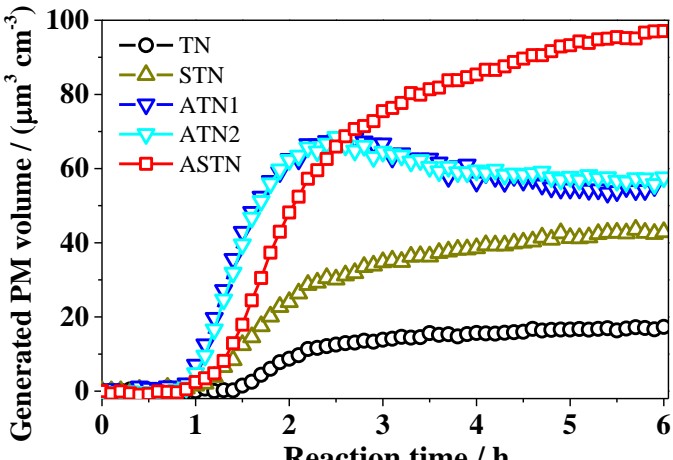


Fig. 1    Secondary aerosol formation in photooxidation of toluene/NO$_x$ in the

presence or absence of NH$_3$ and/or SO$_2$. The letters codes for the experiments

indicate the introduced pollutants, i.e. "A" for ammonia, "S" for sulfur dioxide,

"T" for toluene and "N" for nitrogen dioxide. Experimental details are listed in

Table 1.



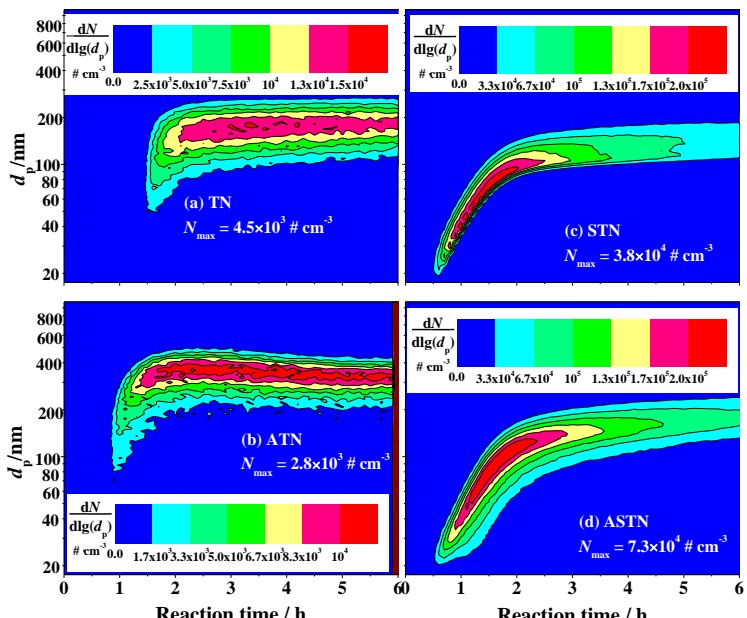

Fig. 2    Size distributions of the suspended particles as a function of time during the

reaction in photooxidation of toluene/$NO_x$ in the presence or absence of $NH_3$

and/or $SO_2$. $N_{max}$ shows the maximal particle number concentration during the

reaction for each experiment. Experimental details are listed in Table 1.

The size distributions of the secondary aerosol in the photooxidation,

with a range of 17-1000 nm, were analyzed and are shown in Fig. 2. A

significant increase in new particle formation was observed in the presence

of $SO_2$. The maximal particle number concentrations in experiments ASTN

and STN were one order of magnitude higher than those in experiments

ATN and TN. The presence of $NH_3$ also contributed substantially to the

particle growth in photooxidation of toluene/$NO_x$. Comparing Fig. 2(c) to

Fig. 2(a), the total number concentration of particles in experiment ATN

was a little lower than that in experiment TN, but the mode diameter of the



particles was much larger.
**Secondary inorganic aerosol formation**
Some synergetic effects were observed in secondary inorganic aerosol
formation besides the generation of ammonium and sulfate from $NH_3$ and
$SO_2$. For example, nitrate formation was not only enhanced by $NH_3$, due to
conversion of nitric acid into ammonia nitrate, but also was markedly
affected by $SO_2$. The chemical compositions of the generated aerosols in
the photooxidation of toluene/$NO_x$ were analyzed with an ACSM, and their
time variations are displayed in Fig. 3. In experiment ATN, the
concentrations of ammonium and nitrate decreased after about 2.3 hours of
reaction, as shown in Fig. 3, which was consistent with the decreasing trend
of particle concentration shown in Fig. 1. The reason for this phenomenon
is unknown but we speculate that the generated $NH_4NO_3$ might partition
back into the gas phase as reaction goes on. In Fig. 2, we observed that the
particle size was larger in experiment ATN than the other three experiments.
The larger diameter resulted in more significant wall deposition, reduced
the surface area of the suspended particles, and shifted the partition
equilibrium to the gas phase. Adding $SO_2$ to the system resulted in a lower
peak concentration but a higher final concentration of nitrate. In the
presence of $SO_2$, higher concentrations of sulfate and organic species were
generated and mixed with nitrate in the aerosol, which may shift the
partition balance of $NH_4NO_3$ to the aerosol phase. In addition, the presence
of organic matter might accelerate the deliquescence of generated



inorganic particles (Meyer et al., 2009;Li et al., 2014), and provide moist
surfaces for heterogeneous hydrolysis of $N_2O_5$, contributing to nitrate
formation (Pathak et al., 2009).

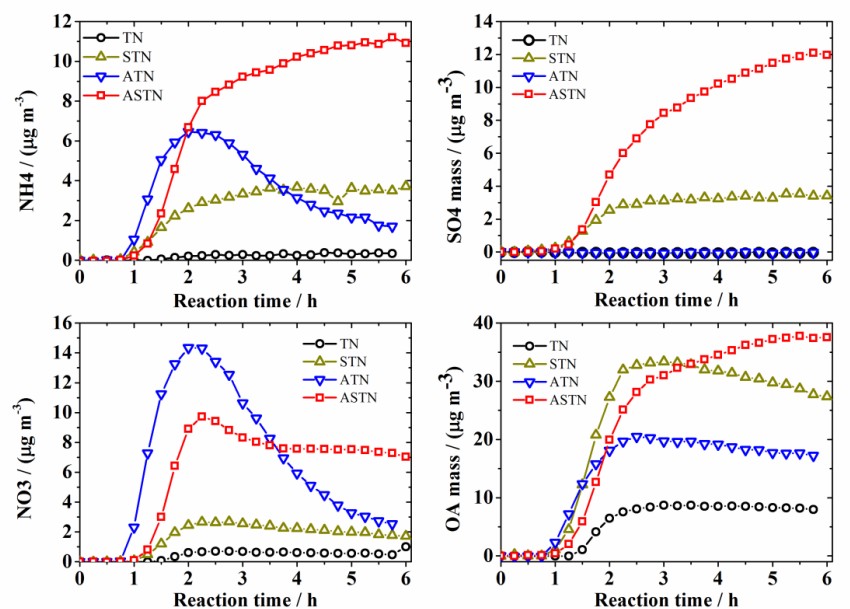


Fig. 3    Time variations of the chemical species in the secondary aerosol generated

from the photooxidation of toluene/$NO_x$ in the presence or absence of $NH_3$ and

$SO_2$. Letter codes for experiments indicate the introduced pollutants, i.e. "A" for

ammonia, "S" for sulfur dioxide, "T" for toluene and "N" for nitrogen dioxide.

Experimental details are listed in Table 1.

In Fig. 3, the generation of ammonium salt can be observed in the

photooxidation of toluene/$NO_x$/$SO_2$ without introducing $NH_3$ gas. This
indicated there was $NH_3$ present in the background air in the chamber, and
also indicated that the effects of $NH_3$ on secondary aerosol formation might
be underestimated in this study. The background $NH_3$ was derived from the





partitioning of the deposited ammonium sulfate and nitrate on the chamber
wall when humid air was introduced (Liu et al., 2015b). Unfortunately, due
to the lack of appropriate instrumentation, we were not able to measure the
exact concentration of $NH_3$ in the background air in the chamber. With this
in mind, the experiments carried out without introducing $NH_3$ gas were
considered "$NH_3$-poor" experiments in this study, while experiments with
the introduction of $NH_3$ gas were considered "$NH_3$-rich" experiments, in
which the concentrations of $NH_3$ were more than twice the $SO_2$
concentrations and the oxidation products of $SO_2$ and $NO_x$ were fully
neutralized by $NH_3$.

To further quantify the effect of $SO_2$ on secondary aerosol formation,

different concentrations of $SO_2$ were introduced under $NH_3$-poor and $NH_3$-
rich conditions. The details of the experimental conditions are shown in
Table 2. In these experiments, the concentrations of toluene were reduced
compared to the experiments in Table 1, and monodisperse $Al_2O_3$ seed
particles with mode diameter about 100 nm were introduced into the
chamber. As shown in Fig. 4, similar to the seed-free experiments, the
presence of $SO_2$ and $NH_3$ clearly increased secondary aerosol formation in
toluene/$NO_x$ photooxidation in the presence of $Al_2O_3$ seed aerosols. In the
experiments carried out in the presence of $Al_2O_3$ seed aerosols, the
decrease of $NH_4NO_3$ was not obvious in the experiment carried out in the
absence of $SO_2$ under $NH_3$-rich conditions, indicating that generation of
$NH_4NO_3$ was highly dependent on the surface area concentration of the
particles, which decreased the partitioning of $NH_4NO_3$ back to the gas



phase, as discussed above.

Under both NH$_3$-poor and NH$_3$-rich conditions, all the detected

chemical species in the generated aerosol, including sulfate, organic
aerosol, nitrate and ammonium, increased linearly with increasing SO$_2$
concentrations, as shown in Fig. 5. The increase was more significant in a
NH$_3$-rich environment than that under NH$_3$-poor conditions, indicating a
synergistic effect of SO$_2$ and NH$_3$ on aerosol generation. Among the four
chemical species, nitrate generation increased most significantly with
respect to SO$_2$ concentration under NH$_3$-rich conditions, followed by
ammonium and organic aerosol, while sulfate was the least sensitive
species. Under NH$_3$-poor conditions, the sensitivity of these species
followed a different sequence, in which sulfate > nitrate > organic aerosol >
ammonium. A better correlation was found between secondary aerosol
formation and particle surface area than that with particle volume, with
details introduced in Fig. S1 in the supporting information, indicating an
enhancement effect in the heterogeneous process rather than in bulk
reactions. The different sequences under NH$_3$-rich and NH$_3$-poor
conditions indicated that the presence of SO$_2$ and NH$_3$ not only contributed
aerosol surface for partitioning, but also enhanced the heterogeneous
process for secondary aerosol formation.

Table 2 Experimental conditions of the toluene/NO$_x$ photooxidation in the presence
of different concentrations of SO$_2$ and Al$_2$O$_3$ seed particles under NH$_3$-poor and NH$_3$-

rich conditions



|  | Toluene$_0$ | NO$_0$ | NO$_x$-NO | SO$_2$ | Al$_2$O$_3$ | NH$_3$* | RH | T |
|---|---|---|---|---|---|---|---|---|
|  | *ppb* | *ppb* | *ppb* | *ppb* | *particle/cm$^3$* | *ppb* | *%* | *K* |
|  | 188 | 147 | 60 | 0 | 2400 | 0 | 50 | 303 |
| NH$_3$-poor | 200 | 126 | 51 | 52 | 3100 | 0 | 50 | 303 |
|  | 188 | 130 | 58 | 105 | 2100 | 0 | 50 | 303 |
|  | 197 | 142 | 46 | 0 | 3300 | 105 | 50 | 303 |
| NH$_3$-rich | 220 | 147 | 50 | 26 | 3300 | 105 | 50 | 303 |
|  | 207 | 145 | 49 | 52 | 3200 | 105 | 50 | 303 |

Calculated according to the introduced amount of NH$_3$ and the volume of the reactor.

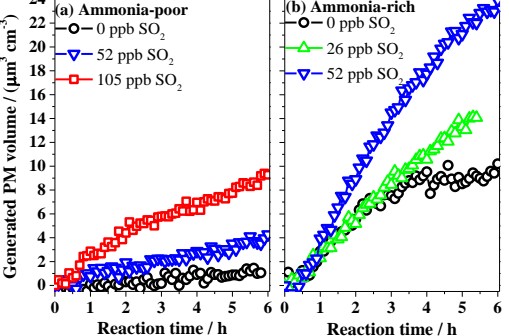


Fig. 4    Secondary aerosol formation as a function of time with different
concentrations of SO$_2$ in the photooxidation of toluene/NO$_x$ under NH$_3$-poor (a)

and NH$_3$-rich (b) conditions. Experimental details are listed in Table 1.





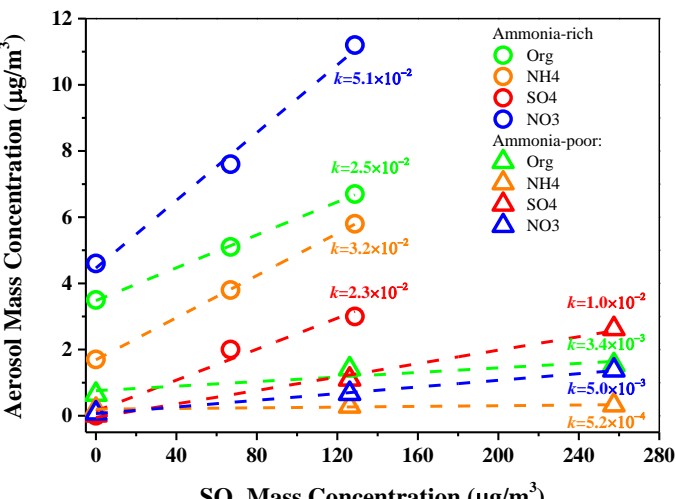

Fig. 5    Formation of nitrate (blue), organic aerosol (green), sulfate (red), and

ammonium salt (orange) as functions of $SO_2$ concentration in the photooxidation

of toluene/$NO_x$ under $NH_3$-rich (circles) or $NH_3$-poor (triangles) conditions. The $k$

values are the slopes of the fitted lines for each species. Experimental details are

listed in Table 1.

Another synergetic effect we found in secondary inorganic aerosol

formation was that sulfate formation was enhanced by the presence of $NH_3$.

In both seed-free experiments and experiments in the presence of $Al_2O_3$

seed aerosols, the sulfate mass concentration was more than tripled under

$NH_3$-rich conditions compared to an $NH_3$-poor environment. This is

consistent with previous studies on the reactions of $SO_2$, $NO_2$ and $NH_3$ in

smog chambers (Behera and Sharma, 2011) and the heterogeneous reaction

between $NH_3$ and $SO_2$ on particle surfaces (Yang et al., 2016; Tursic et al.,



2004). According to the consumption of toluene, OH concentrations in the
photooxidation experiments were estimated to range from $1.6 \times 10^6$
molecules/cm$^3$ to $2.7 \times 10^6$ molecules/cm$^3$. The reaction between these OH
radicals and SO$_2$ contributed 35%-50% of the total SO$_2$ degradation in
NH$_3$-poor experiments, while this ratio was reduced to 25%-30% in NH$_3$-
rich experiments. This indicated that the heterogeneous process was an
important pathway for inorganic aerosol formation in the photooxidation
system, and the heterogeneous process was enhanced by the presence of
NH$_3$. This result is consistent with the finding that failure to include the
heterogeneous process in the model caused an underestimation of SO$_2$
decay in the chamber (Santiago et al., 2012). According to previous studies,
NH$_3$ might provide surface Lewis basicity and liquid surface layers for SO$_2$
absorption and subsequent oxidation, and therefore, enhance sulfate
formation (Yang et al., 2016; Tursic et al., 2004).
**Secondary organic aerosol formation**

The presence of NH$_3$ and SO$_2$ caused significant formation of

secondary inorganic aerosol, and meanwhile, enhanced SOA formation. In
previous studies, Kleindienst et al. (2006) found that the presence of SO$_2$
did not disturb the dynamic reaction system of $\alpha$-pinene or isoprene in the
presence of NO$_x$. In the present study, no obvious difference was found in
the OH concentration in experiments with different concentrations of SO$_2$
and NH$_3$. Therefore, it could be also assumed that the presence of SO$_2$ and
NH$_3$ in this study did not significantly impact the gas phase oxidation of




hydrocarbons and mainly played a role in the aerosol phase.
The presence of $NH_3$ markedly increased aerosol formation in the
photooxidation of toluene/$NO_x$. In the seed-free toluene/$NO_x$
photooxidation experiments, the presence of $NH_3$ caused similar additional
amounts of organic aerosol mass and resulted in increases of 116% and 36%
in the absence or presence of $SO_2$, respectively. In the experiments carried
out in the presence of $Al_2O_3$ seed aerosols, the increase caused by $NH_3$ was
more significant, with the organic aerosol quantity increasing by a factor
of four to five. $NH_3$ may react with the ring opening oxycarboxylic acids
from toluene (Jang and Kamens, 2001), resulting in products with lower
volatility. The presence of $NH_3$ might also change the surface properties of
the aerosol and enhance heterogeneous oxidation of organic products. As
mentioned earlier in this study, there was $NH_3$ present in the background
air in the chamber, so the effects of $NH_3$ on secondary aerosol formation
might be underestimated in this study. Detecting the concentration of $NH_3$
gas as a function of time and quantifying the effects of $NH_3$ on secondary
aerosol are meaningful, and are expected to be studied in the future.
The enhancing effect of $NH_3$ on secondary aerosol formation in toluene
photooxidation was further attributed to its influence in heterogeneous
reactions. In the presence of $Al_2O_3$ seed particles, no obvious new particle
formation was detected in experiments without $SO_2$, as shown in Fig. 6(a)
and Fig. 6(c). The presence of $NH_3$ caused a more noticeable particle
growth of the $Al_2O_3$ seed particles. The increase mainly took place after
0.5 hours of irradiation, and lasted for about an hour, with an average





diameter growth of about 12 nm. In the two experiments carried out in the
presence of 52 ppb SO$_2$ in Fig. 7(b) and Fig. 7(d), significant but similar
new particle formation occurred. The maximum particle number
concentrations detected by the SMPS were about 33000 particle/cm$^3$ and
34000 particle/cm$^3$ under NH$_3$-poor and NH$_3$-rich conditions, respectively.
However, the growth of the seed aerosol in these two experiments was
quite different. Under an NH$_3$-poor condition, the mode diameter of the
seed aerosols grew from 100 nm to about 130 nm, while under an NH-rich
condition it grew to about 220 nm. These results indicated that elevated
NH$_3$ concentrations mainly affected secondary aerosol formation in the
heterogeneous process.

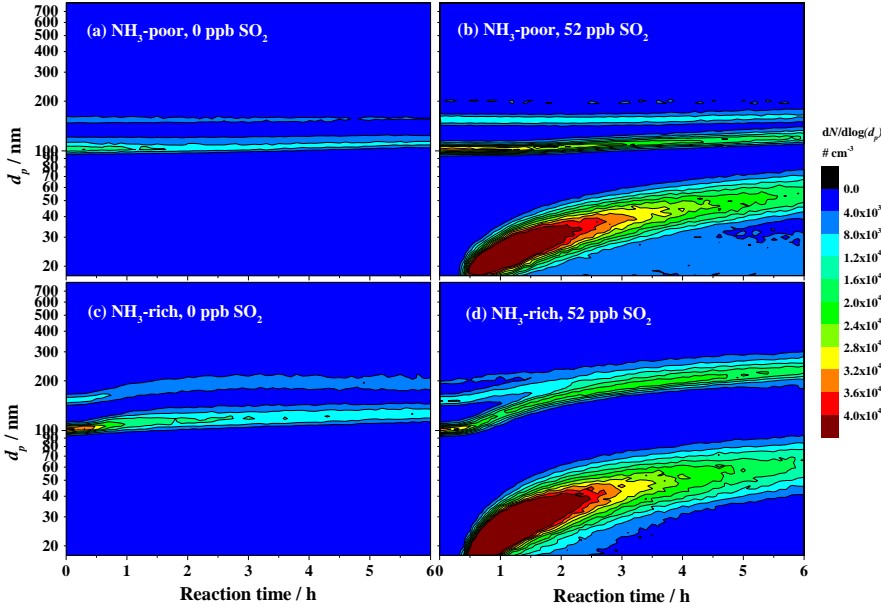


Fig. 6    Size distributions of the suspended particles as a function of time during the

reaction in photooxidation of toluene/NO$_x$ in the presence of Al$_2$O$_3$ seed particles.





Experimental details are listed in Table 1.


The chemical properties of the generated SOA under different
conditions of $NH_3$ and $SO_2$ were compared by applying PMF analysis to
the AMS data. Two factors were identified from the analysis, with average
elemental composition of $CH_{0.82}O_{0.75}N_{0.051}S_{0.0014}$ for Factor 1 and
$CH_{1.05}O_{0.55}N_{0.039}S_{0.0017}$ for Factor 2. The difference mass spectra between
the two factors are shown in Fig. 7. The abundance of $C_xH_y$ fragments was
higher in Factor 2 than Factor 1, while oxygen and nitrogen content in
Factor 1 were higher than Factor 2. Meanwhile, as indicated in the red box
in Fig. 7, fragments with high $m/z$ were more abundant in Factor 2. Thus
we assigned Factor 1 to the highly oxidized organic component and some
nitrogenous organic compounds, while Factor 2 was assigned to less-
oxidized organic aerosol and some oligomers.




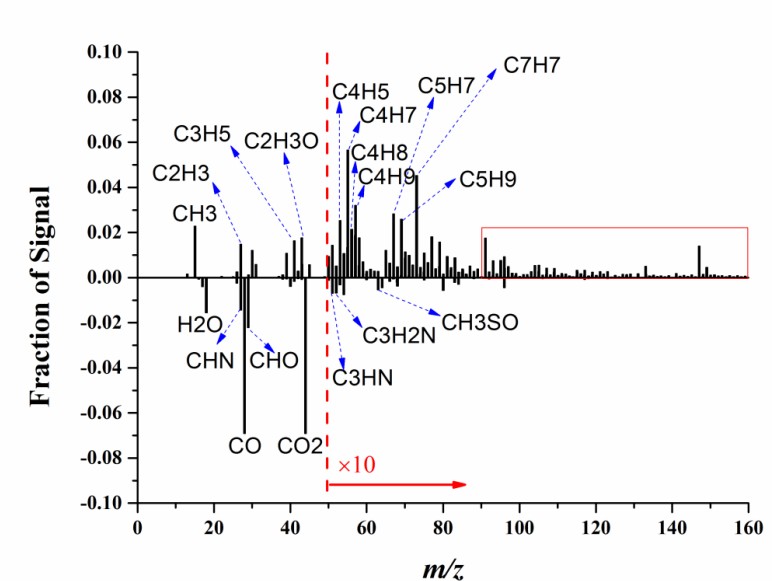


Fig. 7    The difference mass spectra (Factor 2 – Factor 1) between the two factors of

the generated organic aerosol identified by applying PMF analysis to the AMS

data

These two factors had different temporal variations during the reaction.
As indicated in Fig. 8, Factor 2 always increased at the beginning of the
reaction but decreased after reaching a peak with 1 or 2 hours of irradiation.
Factor 1 was generated later than Factor 2, while it continuously increased
during the reaction. Comparing experiments with different concentrations
of $SO_2$, the production of Factor 2 increased with increasing $SO_2$ under
$NH_3$-poor conditions, while Factor 1 increased with increasing $SO_2$ under
an $NH_3$-rich environment. Similar results can also be found in Fig. 9. The
higher production of Factor 2 with higher $SO_2$ under an $NH_3$-poor
environment could be probably attributed to the well-known acid-catalysis





effects of the oxidation product of $SO_2$, i.e. sulfuric acid, on heterogeneous
aldol condensation (Offenberg et al., 2009; Jang et al., 2002; Gao et al.,
2004). Under $NH_3$-rich conditions, however, Factor 1, which has higher
contents of oxygen and nitrogen than Factor 2, dominated in the SOA
formation. Meanwhile, the production of Factor 2 increased significantly
with increasing $SO_2$ concentration in $NH_3$-rich conditions. This indicated
that the formation of highly oxidized organic compounds and nitrogenous
organic compounds was increased with higher concentrations of $SO_2$ under
$NH_3$-rich conditions. By inference and from the results of AMS
measurements, aerosol water increased as the initial concentration of $SO_2$
increased, since more inorganic aerosol was generated. Liggio and Li
(2013) suggest that dissolution of primary polar gases into a partially
aqueous aerosol contributed to the increase of organic mass and oxygen
content on neutral and near-neutral seed aerosols, which would also take
place in the $NH_3$-rich experiments and contribute to the generation of
Factor 1.





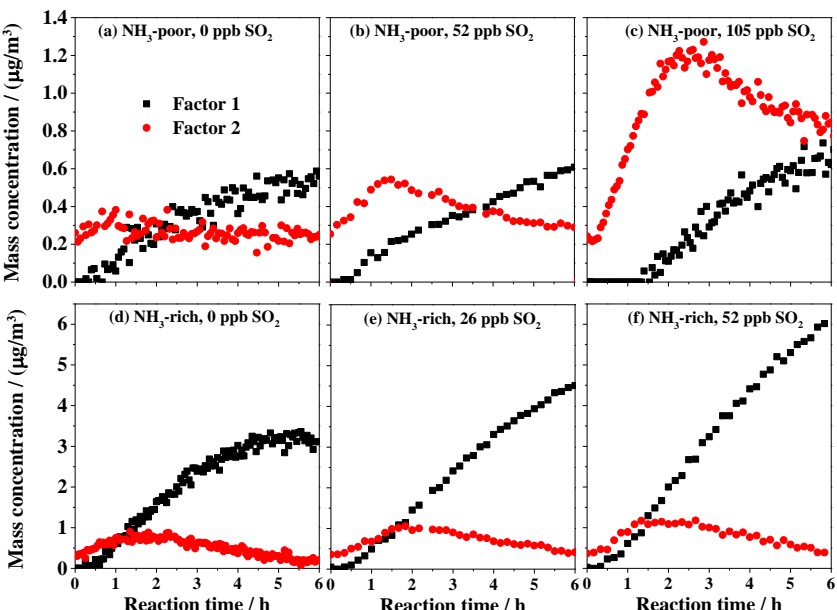

Fig. 8    Temporal variations of Factor 1 and Factor 2 in the presence of different

concentrations of $SO_2$ under $NH_3$-poor and $NH_3$-rich conditions.

Nitrogen-containing organics (NOC) are a potentially important aspect

of SOA formation, and may have contributed to the increase of Factor 1 in

this study. NOC might contain organonitrates, formed through reactions

between organic peroxy radicals ($RO_2$) and NO (Arey et al., 2001), organic

ammonium    salts,    generated    in    acid-base    reactions    between

ammonia/ammonium and organic acid species (Liu et al., 2012b), and

species with carbon covalently bonded to nitrogen, generated in reactions

of ammonia/ammonium with carbonyl functional group organics (Wang et

al., 2010). Although we were not able to measure NOC, some indirect

estimation methods suggested by Farmer et al. (2010) could be applied.

The details for estimation of the concentrations of organonitrates and NOC





with reduced N are given in the supporting information. Despite the
uncertainty, there is an obvious increasing trend of organonitrates and NOC
with reduced N with increasing $SO_2$ concentration under $NH_3$-rich
conditions, as shown in Fig. 9. The increase ratio of NOC is higher than
that of the organic aerosol or Factor 1 as $SO_2$ concentration increases. The
estimated NOC contributed most of the increase in Factor 1 in $NH_3$-rich
conditions. These results provide some evidence that the formation of
organonitrates and NOC with reduced N (organic ammonium salts, imines,
imidazole, and so on) played an important role in the increasing trend of
SOA with $SO_2$ in a $NH_3$-rich environment. It was speculated that the higher
surface acidity of aerosol formed in the presence of a high concentration of
$SO_2$ favors NOC formation through $NH_3$ uptake by SOA, as observed in a
recent work (Liu et al., 2015b).

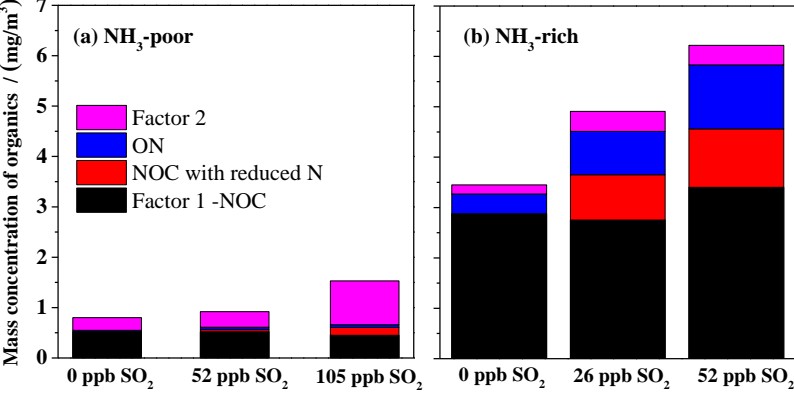


Fig. 9   The estimated concentrations of NOC (ON+NOC with reduced N)

and the two factors (identified by PMF analysis) in SOA as a function

of $SO_2$ concentration in photooxidation of toluene/$NO_x$ under (a)





467           NH$_3$-poor and (b) NH$_3$-rich conditions

## Conclusions

In the photooxidation system of toluene/NO$_x$, the presence of SO$_2$
and/or NH$_3$ increased secondary aerosol formation markedly, regardless of
whether Al$_2$O$_3$ seed aerosol was present or not. Some synergetic effects in
the heterogeneous process were observed in secondary inorganic aerosol
formation in addition to the generation of ammonium and sulfate from NH$_3$
and SO$_2$. Specifically, the generation of NH$_4$NO$_3$ was found to be highly
dependent on the surface area concentration of suspended particles, and
was enhanced by increased SO$_2$ concentration. Meanwhile, sulfate
formation was also increased in the presence of NH$_3$. The absorbed NH$_3$
might provide liquid surface layers for the absorption and subsequent
reaction for SO$_2$ and organic products, and therefore, enhance sulfate and
SOA formation. NH$_3$ mainly influenced secondary aerosol formation in the
heterogeneous process, resulting in significant growth of seed aerosols, but
had little influence on new particle generation. In the experiments carried
out in the presence of Al$_2$O$_3$ seed aerosols, sulfate, organic aerosol, nitrate
and ammonium were all found to increase linearly with increasing SO$_2$
concentrations in toluene/NO$_x$ photooxidation. The increase of these four
species was more obvious under NH$_3$-rich conditions, and the order of their
sensitivity was different from that under NH$_3$-poor conditions. A better
correlation between secondary aerosol formation and particle surface area
than that with particle volume indicated an enhancement effect in the





heterogeneous process rather than in bulk reactions.
Two factors were identified in the PMF analysis of the AMS data. One
factor assigned to less-oxidized organic aerosol and some oligomers
increased with increasing $SO_2$ under $NH_3$-poor conditions, mainly due to
the well-known acid catalytic effects of the acid products on SOA
formation in the heterogeneous process. The other factor, assigned to the
highly oxidized organic component and some nitrogenous organic
compounds, increased with increasing $SO_2$ under an $NH_3$-rich environment,
with NOC (organonitrates and NOC with reduced N) contributing most of
the increase.
This study indicated that the synergistic effects between inorganic
pollutants could substantially enhance secondary inorganic aerosol
formation. Meanwhile, the presence of inorganic gas pollutants, i.e. $SO_2$
and $NH_3$, promoted SOA formation markedly. Synergistic formation of
secondary inorganic and organic aerosol might increase the secondary
aerosol load in the atmosphere. These synergistic effects were related to
the heterogeneous process on the aerosol surface, and need to be quantified
and considered in air quality models.

## Acknowledgments


This work was supported by the National Natural Science Foundation of
China (21407158), the "Strategic Priority Research Program" of the
Chinese Academy of Sciences (XDB05010300, XDB05040100,
XDB05010102), and the special fund of the State Key Joint Laboratory of





Environment Simulation and Pollution Control (14Z04ESPCR). This work
was also financially and technically supported by Toyota Motor
Corporation and Toyota Central Research and Development Laboratories
Inc.

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
