# Peer review of "Synergetic formation of secondary inorganic and organic aerosol: Influence of SO₂ and/or NH₃ in the heterogeneous process"

_Atmospheric Chemistry and Physics, 2016_

## Referee Comment (RC1) · Anonymous Referee #1 · 3 Aug 2016

The authors investigated the effect of sulphur dioxide and ammonia on the secondary organic aerosol formation in the photooxidation of toluene/NOx with or without $Al_2O_3$ seed particles. They presented new experimental results and showed that the synergistic effects between $SO_2$ and $NH_3$ in the heterogeneous process can greatly enhance the aerosol formation. While the authors presented valuable experimental data, they should provide detailed explanations to support their arguments and observations made in the manuscript.

Comments

One main question is:  How much toluene is being reacted under different experimental conditions? This information is not given in the manuscript. If the amount of the toluene reacted is known, what are the aerosol mass yields in these experiments? How do the measured aerosol mass yields compare with literature results, if any?

Line 184, "Assuming the same aerosol density in these experiments, the presence of either $NH_3$ or $SO_2$ enhanced secondary aerosol formation markedly" Can the authors provide justification for this assumption?

Line 217, "A significant increase in new particle formation was observed in the presence of $SO_2$." Can the authors provide an explanation for this observation?

 Line 232, "In experiment ATN, the concentrations of ammonium and nitrate decreased after about 2.3 hours of reaction, as shown in Fig. 3, which was consistent with the decreasing trend of particle concentration shown in Fig. 1. The reason for this phenomenon is unknown but we speculate that the generated $NH_4NO_3$ might partition back into the gas phase as reaction goes on." Can the authors preform any calculations or model simulations to support this hypothesis? What the concentration of the $NH_3$, $SO_2$ and $NO_x$ as a function of time in these experiments?

Line 245, "In addition, the presence of organic matter might accelerate the deliquescence of generated inorganic particles (Meyer et al., 2009;Li et al., 2014), and provide moist surfaces for heterogeneous hydrolysis of N2O5, contributing to nitrate formation (Pathak et al., 2009)." What is the meaning of "the deliquescence of generated inorganic particles"? Please elaborate. Does the $N_2O_5$ form under the experimental conditions?

Line 256, "In Fig. 3, the generation of ammonium salt can be observed in the photooxidation of toluene/$NO_x$/$SO_2$ without introducing $NH_3$ gas. This indicated there was $NH_3$ present in the background air in the chamber, and also indicated that the effects of $NH_3$ on secondary aerosol formation might be underestimated in this study." As stated, in order to better access the impact of $NH_3$ on the aerosol formation, it is important to know the background $NH_3$ concentration. The authors should give a reasonable guess or estimate on the background $NH_3$ concentration in their experiments. Could the authors estimate background $NH_3$ concentration based on their experimental results (e.g. STN data)?

Line 267, "in which the concentrations of $NH_3$ were more than twice the $SO_2$ concentrations and the oxidation products of $SO_2$ and $NO_x$ were fully neutralized by $NH_3$." Any experimental data or calculations to support this argument.

Line 274, "In these experiments, the concentrations of toluene were reduced compared to the experiments in Table 1, and monodisperse $Al_2O_3$ seed particles with mode diameter about 100 nm were introduced into the chamber." Any reason why a lower concentration of toluene is used for the $Al_2O_3$ seed particle experiments.

Line 279, "In the experiments carried out in the presence of $Al_2O_3$ seed aerosols, the decrease of $NH_4NO_3$ was not obvious in the experiment carried out in the absence of $SO_2$ under $NH_3$-rich conditions, indicating that generation of $NH_4NO_3$ was highly dependent on the surface area concentration of the particles, which decreased the partitioning of $NH_4NO_3$ back to the gas phase, as discussed above". In addition to describe the results in text, can the authors show the time variation of the chemical species in the SOA generated from the toluene oxidation in the presence of $Al_2O_3$ seed aerosols to support their argument?

Line 297, "A better correlation was found between secondary aerosol formation and particle surface area than that with particle volume, with details introduced in Fig. S1 in the supporting information, indicating an enhancement effect in the heterogeneous process rather than in bulk reactions". From the Figure S1, the authors argue that there is a significant correlation between the aerosol formation and aerosol surface area. From that point, they suggest that the heterogeneous processes is important than the bulk reactions. However, a strong conclusion cannot be easily drawn from the simple correlation method and limited data points. Furthermore, a nice correlation between the aerosol formation and aerosol volume is observed. Can the authors further elaborate this point?

Line 330, "According to the consumption of toluene, OH concentrations in the photooxidation experiments were estimated to range from $1.6x10^6$ molecules/$cm^3$ to $2.7 x10^6$ molecules/$cm^3$. The reaction between these OH radicals and $SO_2$ contributed 35%-50% of the total $SO_2$ degradation in $NH_3$-poor experiments, while this ratio was reduced to 25%-30% in $NH_3$-rich experiments" Again, what is the amount of toluene and $SO_2$ reacted in the experiments? What the time variations of the toluene and $SO_2$ and other gas phase species (e.g. $NO_x$) in these experiments?

Line 341, "According to previous studies, $NH_3$ might provide surface Lewis basicity and liquid surface layers for $SO_2$ absorption and subsequent oxidation, and therefore, enhance sulfate formation (Yang et al., 2016; Tursic et al., 2004)." What is the physical state of the aerosols in the experiments? Aqueous droplets or solid particles?

Line 416, "The higher production of Factor 2 with higher $SO_2$ under an $NH_3$-poor environment could be probably attributed to the well-known acid-catalysis effects of the oxidation product of $SO_2$, i.e. sulfuric acid, on heterogeneous aldol condensation (Offenberg et al., 2009; Jang et al., 2002; Gao et al., 2004)" What is the pH or acidity of the aerosols in these experiments?

Line 427 "By inference and from the results of AMS measurements, aerosol water increased as the initial concentration of $SO_2$ increased, since more inorganic aerosol was generated. Liggio and Li (2013) suggest that dissolution of primary polar gases into a partially aqueous aerosol contributed to the increase of organic mass and oxygen content on neutral and near-neutral seed aerosols, which would also take place in the $NH_3$-rich experiments and contribute to the generation of Factor 1." What is the amount of aerosol phase water and how the aerosol water content change with reactions in these experiments?

---

## Referee Comment (RC2) · Anonymous Referee #2 · 21 Sep 2016

The manuscript reports data related to the effect of SO2 and NH3 on aerosol formation from the oxidation of toluene in the presence on NOx. The experimental study was conducted in the presence or absence of inorganic seed aerosol: AL2O3. NH3 and SO2 are two species emitted into the atmosphere and can have a large effect on atmospheric chemistry. The data analysis show aerosol formation and growth increased in the presence of SO2 regardless of the presence of NH3. This study and its topic is of great interest and appropriate to ACPD journal. This study is worth to be published since it present an important set of data that can be useful for the atmospheric communities. However, I feel that the text and the scientific discussion (interpretation of experimental data) (see my comments below) need to be addressed before publication.

[Figure]

The experimental part needs to be addressed and clearly state how the experiments were run. Toluene as well other gas phase species need to be reported vs time in this study. The role of OH radicals should be discussed? The wall loss of gas phase and particles should be addressed also? The errors and uncertainties need to be addressed since assumptions were made in this study. Yield should be reported in this study for the different systems studied.

As I mentioned, the manuscript reports a set of great data important to scientist interested in atmospheric organic and inorganic aerosol formation and the effect of NH3 and SO2!

Comments:

In the introduction (1st paragraph), the authors report literature data for NH3 in China and almost no data was provided for SO2. I suggest SO2 should be provided also and a comparison should be reported between SO2 and NH3. The text in the manuscript should be edited for consistency. I found it very hard to follow the authors' ideas in the manuscript, although lot of information is provided. For example, sentences reported between lines 89 and 115 are very difficult to follow for me!!! This is true for most the manuscript!

Line 125: this study focusses on "toluene" and "VOC" should be deleted.

The chamber was a 2 m3 and the losses expected to be higher. The authors should give more information in this study about the wall losses of gas phase and particles.

How NH3 was estimated (line 168)? Needs errors and uncertainties?

Line 171. title not clear to me? Be specific I would suggest: Effect of NH3 and SO2 on particle formation and growth

Table 1 change "hydrocarbon" to "toluene"

Data provided in Table 1 are initial concentration? The authors need to specify how the

chamber was run (as flow reactor or batch reactor). Based on fig 1, it seems to me it was conducted as a batch reactor. In this case the wall losses are important and need to be incorporated in the discussion and how toluene and SMPS data were analyzed to get to Figs 1, 2.. and tables 1, 2.

It is also very important to provide data for Toluene reacted, $SO_2$ reacted, $NH_3$ reacted and NO reacted, $NO_x$ reacted (vs. time) in the gas phase in a separate figure.

The experiments were conducted under 50% RH. The authors didn't reported in the text until Table 1 was mentioned. This is a very important parameters that should be reported and discussed at list briefly. How it was measured and controlled in the chamber!!! Same thing for the temperature! I'm expecting the RH will change aver the run time and will be not constant?

How $NH_3$ was introduced into the chamber? Please elaborate!

It's important to have the amount of toluene reacted in each case in order to see how much was oxidized. then measure the yield etc... The OH radicals present in the system can be reacted with different gas phase species (e.g. toluene, $SO_2$,..) and then depending on the rate constant, it can affect the conclusion reported in this study.

Line 225...How the authors distinguish between secondary organic aerosol and secondary inorganic aerosol in these experiments?

Line 228-230 How the authors come to this conclusion? It's speculation and not based on data reported here. How nitrate are measured in this study? Are you refering to inorganic or organic nitrates? It will be great if a distinction was made between SOA and secondary inorganic aerosol in this study?

It will be interesting to estimate the concentration of OH radicals vs time in these experiments? Line 239. "The larger diameter resulted in more significant wall deposition, reduced the surface area of the suspended particles, and shifted the partition equilibrium to the gas phase." Are the authors measured the wall losses at different size

distribution or this only speculation?

Line 245 – 249. It seems to me that these were not based on data reported in these study. How N2O5 plays a role here? N2O5 was measured in this study? How it was formed in the chamber. Is ozone was measured in this study? If yes it should be reported vs time.

Lines 256. Is ammonia salt was measured? How the authors come to the measured ammonia salt.

NH3 was estimated in the chamber according to Table 2. Why NH3 was not measured experimentally vs time? This is very important (same for SO2). I suggest to use "experiment without NH3 added to the chamber" instead of "NH3 poor"? It's confusing and make it difficult for comparison with other data?

Line 314. Should be Table 2.

---

## Author Comment (AC1) · 31 Oct 2016

**Response for Reviewer #1**

**Ms. Ref. No.: acp-2016-486**

**Title: "Synergetic formation of secondary inorganic and organic aerosol: Influence of $SO_2$ and/or $NH_3$ in the heterogeneous process"**

**Revised Title: "Synergetic formation of secondary inorganic and organic aerosol: Effect of $SO_2$ and $NH_3$ on particle formation and growth"**

We appreciate the comments from the reviewer on this manuscript. We have answered them in the following paragraphs (the text in italics is the reviewer comments, followed by our response) point by point. The text in blue is some revisions for the manuscript. The line numbers in the response are from the revised manuscript.

*The authors investigated the effect of sulphur dioxide and ammonia on the secondary organic aerosol formation in the photooxidation of toluene/NOx with or without $Al_2O_3$ seed particles. They presented new experimental results and showed that the synergistic effects between $SO_2$ and $NH_3$ in the heterogeneous process can greatly enhance the aerosol formation. While the authors presented valuable experimental data, they should provide detailed explanations to support their arguments and observations made in the manuscript.*

*Comments*

*One main question is: How much toluene is being reacted under different experimental conditions? This information is not given in the manuscript. If the amount of the toluene reacted is known, what are the aerosol mass yields in these experiments? How do the measured aerosol mass yields compare with literature results, if any?*

**Response:** Thanks for the reviewer's comments. Time variations of gas-phase compounds in photooxidation of toluene/$NO_x$ in the presence or absence of $NH_3$

and/or $SO_2$ are displayed in Figure R1 and Figure R2. These Figures has been added in the revised Supporting information. The reacted amount of toluene was calculated and has been added in the revised manuscript. The presence of $SO_2/NH_3$ had no obvious effect on the reacted amount of toluene, as shown in Table R1 and Table R2. SOA yields were also calculated. In Figure R3, the SOA yields in this study are compared with literature results. SOA yields in in photooxidation of toluene/$NO_x$ in the presence or absence of $NH_3$ and/or $SO_2$ were similar as that reported by Odum, J. R., et. al.. A closer inspection revealed that experiment TN had SOA yield a little lower than the curve in the study of Odum, J. R., et. al., experiment STN and ATN had SOA yields quite close to the curve, while experiment ASTN had a little higher yield than the curve. For SOA yields in photooxidation of toluene/$NO_x$ with different concentrations of $SO_2$, SOA yields were higher in $NH_3$-rich condition compare to $NH_3$-poor condition. And there is a trend that SOA yield increased with increasing $SO_2$ concentrations. The presence of $SO_2/NH_3$ increased SOA yield.

**Table R1.** The consumption of gas precursors, the formation of ozone and SOA, and SOA yield in photooxidation of toluene/$NO_x$ in the presence or absence of $NH_3$ and/or $SO_2$. The letters codes for the experiments indicate the introduced pollutants, i.e. "A" for ammonia, "S" for sulfur dioxide, "T" for toluene and "N" for nitrogen dioxide.

| Experiment No. | Δtoluene ppm | ΔNO$_x$ ppb | ΔSO$_2$ ppb | ΔO$_3$ ppb | SOA μg/m$^3$ | SOA yield % |
|---|---|---|---|---|---|---|
| TN | 0.19 | 89 | NA | 179 | 8.0 | 1.1 |
| STN | 0.18 | 90 | 25 | 176 | 27.7 | 4.1 |
| ATN | 0.16 | 82 | NA | 159 | 17.2 | 2.9 |
| ASTN | 0.15 | 90 | 42 | 166 | 37.4 | 6.8 |

**Table R2.** The consumption of gas precursors, the formation of ozone and SOA, and SOA yield in photooxidation of toluene/$NO_x$ with different concentrations of $SO_2$ under $NH_3$-poor and $NH_3$-rich conditions.

| | Δtoluene ppb | ΔNO$_x$ ppb | ΔSO$_2$ ppb | ΔO$_3$ ppb | SOA μg/m$^3$ | SOA yield % |
|---|---|---|---|---|---|---|
| NH$_3$-poor | 41 | 73 | NA | 22 | 0.6 | 0.4 |

| | | | | | |
|---|---|---|---|---|---|
| | 40 | 61 | 6 | 22 | 1.4 | 1.0 |
| | 39 | 62 | 12 | 20 | 1.5 | 1.1 |
| | 47 | 79 | NA | 27 | 3.5 | 2.0 |
| NH$_3$-rich | 45 | 81 | 6 | 26 | 5.1 | 3.1 |
| | 44 | 75 | 11 | 27 | 6.7 | 4.1 |

[Figure]

**Figure R1.** Time variations of gas-phase compounds in photooxidation of toluene/NO$_x$ in the presence or absence of NH$_3$ and/or SO$_2$. The letters codes for the experiments indicate the introduced pollutants, i.e. "A" for ammonia, "S" for sulfur dioxide, "T" for toluene and "N" for nitrogen dioxide.

[Figure]

**Figure R2.** Time variations of gas-phase compounds in photooxidation of toluene/NO$_x$ with different concentrations of SO$_2$ under NH$_3$-poor and NH$_3$-rich conditions

[Figure]

**Figure R3.** SOA yields in the experiments in this study and the comparison with literature results (Takekawa et al., 2003;Hildebrandt Ruiz et al., 2015;Odum et al., 1997;Sato et al., 2007)

**Revision in the manuscript:**

**Lines 222-225, Add:** "Time variations of gas phase compounds of these experiments are shown in Fig. S1 in the supporting information. The presence of $SO_2$ and/or $NH_3$ had no obvious effect on the gas phase compounds, including toluene, $NO_x$, $SO_2$ and $O_3$."

**Add Table R1 and Table R2 in the supporting information**

**Add Fig. R1, Fig. R2 and Fig. R3 in the supporting information**

**Lines 428-430, Add:** "The increases of SOA mass in the presence of $NH_3$ and $SO_2$ are shown in Fig. 5. Similar trends for SOA yields can be found in the supporting information."

*Line 184, "Assuming the same aerosol density in these experiments, the presence of either $NH_3$ or $SO_2$ enhanced secondary aerosol formation markedly" Can the authors provide justification for this assumption?*

**Response:** This assumption is actually not true. In the presence of $NH_3/SO_2$, more

inorganic aerosol (higher mass proportion) was generated than the experiment in the absence of $NH_3/SO_2$. The density of inorganic aerosol, mainly sulfate, nitrate and ammonium, is about 1.7 $g/cm^3$, while it is about 1.4 $g/cm^3$ for SOA. Therefore, the assumption of a same aerosol density would under estimate the increase effect of $NH_3$ or $SO_2$ on secondary aerosol formation. To keep things simple and to avoid misunderstanding, this sentence has been deleted in the revised manuscript.

**Revision in the manuscript:**

**Lines 229-231, Delete:** "Assuming the same aerosol density in these experiments, the presence of either $NH_3$ or $SO_2$ enhanced secondary aerosol formation markedly."

*Line 217, "A significant increase in new particle formation was observed in the presence of $SO_2$." Can the authors provide an explanation for this observation?*

**Response:** The new particle formation was not directly measured in this study. The comparison about new particle formation was made based on the number concentrations of particles. In the two experiments in the presence of $SO_2$, the maximum number concentration of particles are $3.8 \times 10^4$ particles/cm$^3$ and $7.3 \times 10^4$ particles/cm$^3$. This was one magnitude higher than that in the experiments in the absence of $SO_2$, in which the maximum number concentration of particles are $4.5 \times 10^3$ particles/cm$^3$ and $2.8 \times 10^3$ particles/cm$^3$, as shown in Fig. 2 in the manuscript. This explanation has been added in the revised manuscript.

**Revision in the manuscript:**

**Lines 261-265, Change:** "A significant increase in new particle formation was observed in the presence of $SO_2$."

**To:** "The new particle formation was not directly measured in this study, but the newly generated particles could be detected when the particles increased in size. According to the particle number concentrations, new particle formation appeared to increase a great deal in the presence of $SO_2$."

*Line 232, "In experiment ATN, the concentrations of ammonium and nitrate decreased after about 2.3 hours of reaction, as shown in Fig. 3, which was consistent*

*with the decreasing trend of particle concentration shown in Fig. 1. The reason for this phenomenon is unknown but we speculate that the generated NH4NO3 might partition back into the gas phase as reaction goes on." Can the authors preform any calculations or model simulations to support this hypothesis? What the concentration of the NH3, SO2 and NOx as a function of time in these experiments?*

**Response:** The partition of $NH_4NO_3$ has been simulated using the AIM Aerosol Thermodynamics Model. The detail of the model is available at http://www.aim.env.uea.ac.uk/aim/aim.php, and was described elsewhere (Clegg and Brimblecombe, 2005;Clegg et al., 1998;Carslaw et al., 1995). The simulation results are summarized in Fig. R4. This figure has been added in the supporting information. The results showed that the concentrations of $NH_3$ gas and coexisted $(NH_4)_2SO_4$ both influenced the partition balance between $NH_4NO_3$ and $HNO_3+NH_3$ in the gas phase. The deposition of $NH_3$ gas and $(NH_4)_2SO_4$ were likely to shift balance to the gas phase and reduce the concentration of $NH_4NO_3$ salt. While the concentration of $NH_4NO_3$ salt seemed not to be affected by the deposition of $NH_4NO_3$, as long as the wall loss was corrected accurately. As we mentioned in the manuscript, the deposition correction was introduced in our previous studies (Chu et al., 2012;Chu et al., 2014). We measured the deposition rates of $(NH_4)_2SO_4$ aerosol with different sizes and fitted them into an empirical function. Then, the deposition of aerosols was corrected based on the function and the size distribution of aerosols measured by the SMPS. According to these results, some revision has been made in the revised manuscript.

The time variations of $NO_x$ and $SO_2$ are shown in Fig. R1 as mentioned earlier. However, the concentrations of $NH_3$ were not measured in this study due to the lack of analytical instruments. We noticed the concentration of $NH_3$ is crucial for a quantitative study about the effects of $NH_3$ on secondary aerosol formation. This problem is expected to be solved in a future study.

[Figure]

**Figure R4.** Concentrations of NH₄NO₃ salt as a function of the wall deposition of NH₃, $(NH_4)_2SO_4$ and $NH_4NO_3$ based on the simulation results of the AIM Aerosol Thermodynamics Model

**Revision in the manuscript:**

**Lines 289-302, Change:** "In 错误!未找到引用源。, we observed that the particle size was larger in experiment ATN than the other three experiments. The larger

diameter resulted in more significant wall deposition, reduced the surface area of the suspended particles, and shifted the partition equilibrium to the gas phase."

**To:** "Detailed simulation results based on the AIM Aerosol Thermodynamics Model (Clegg and Brimblecombe, 2005; Clegg et al., 1998; Carslaw et al., 1995) are shown in Fig. S3 in the supporting information. The deposition of $NH_3$ in the experiment was likely to shift the partition equilibrium to the gas phase and reduce the concentration of $NH_4NO_3$ salt. In addition, the wall deposition of aerosols might also introduce some error in the concentrations of $NH_4NO_3$ salt, although wall deposition was corrected using an empirical function based on deposition rates of $(NH_4)_2SO_4$ aerosol with different sizes (Chu et al., 2012;Chu et al., 2014)."

**Add Fig. R4 in the supporting information**

**Lines 305-308, Add:** "Some simulation results using the AIM Aerosol Thermodynamics Model with different concentrations of sulfate are also shown in Fig. S3 in the supporting information."

*Line 245, "In addition, the presence of organic matter might accelerate the deliquescence of generated inorganic particles (Meyer et al., 2009;Li et al., 2014), and provide moist surfaces for heterogeneous hydrolysis of N2O5, contributing to nitrate formation (Pathak et al., 2009)." What is the meaning of "the deliquescence of generated inorganic particles"? Please elaborate. Does the N2O5 form under the experimental conditions?*

**Response:** In the presence of organic compounds, $(NH_4)_2SO_4$ was reported to deliquesce at RH lower than pure $(NH_4)_2SO_4$. (Meyer et al., 2009;Li et al., 2014). To avoid misunderstanding, the description was revised.

$N_2O_5$ was not measured in this study, but it was expected to be generated in the presence of $NO_2$ and $O_3$ in the experiments. Under the experimental conditions, the maximum formation velocity was calculated to be about 13 ppb/hour from gas phase reaction between $NO_2$ and $O_3$. The concentrations of $NO_2$ and $O_3$ are shown in Fig. R1. The reaction constant was summarized by Atkinson et al. (2004). The uptake coefficient of $N_2O_5$ on particle surface was reported to be about $10^{-2}$ on ammonium

sulfate (Hallquist et al., 2003;Hu and Abbatt, 1997), but would decrease when organics coated on the sulfate (Anttila et al., 2006). The particle surface area concentration in experiment ASTN ranged from 0 to 0.1 $m^2/m^3$. Assuming a concentration of $N_2O_5$ of 0.1 ppb and an uptake coefficient of $10^{-3}$ for $N_2O_5$, the heterogeneous hydrolysis of $N_2O_5$ on 0.05 $m^2/m^3$ suspended particles would generate 6 $\mu g/m^3$ nitrate per hour in the reactor. Thus we speculated the heterogeneous hydrolysis of $N_2O_5$ might be important in the experiment. Some of these explanations have been added in the revised manuscript.

**Revision in the manuscript:**

**Lines 308-317, Change:** "In addition, the presence of organic matter might accelerate the deliquescence of generated inorganic particles (Meyer et al., 2009;Li et al., 2014), and provide moist surfaces for heterogeneous hydrolysis of $N_2O_5$, contributing to nitrate formation (Pathak et al., 2009)."

**To:** "In addition, in the presence of organic matter, $(NH_4)_2SO_4$ aerosol might deliquesce at a RH lower than the deliquescence relative humidity (DRH) (Meyer et al., 2009;Li et al., 2014). If this took place in the experiment, sulfate might provide moist surfaces for heterogeneous hydrolysis of $N_2O_5$, contributing to nitrate formation due to the high uptake coefficient of $N_2O_5$ on ammonium sulfate (Pathak et al., 2009;Hallquist et al., 2003;Hu and Abbatt, 1997). $N_2O_5$ was not measured in this study, but it was expected to be generated in the presence of $NO_2$ and $O_3$ in the experiments."

*Line 256, "In Fig. 3, the generation of ammonium salt can be observed in the photooxidation of toluene/NOx/SO2 without introducing NH3 gas. This indicated there was NH3 present in the background air in the chamber, and also indicated that the effects of NH3 on secondary aerosol formation might be underestimated in this study." As stated, in order to better access the impact of NH3 on the aerosol formation, it is important to know the background NH3 concentration. The authors should give a reasonable guess or estimate on the background NH3 concentration in*

*their experiments. Could the authors estimate background NH3 concentration based on their experimental results (e.g. STN data)?*

**Response:** Thanks for the good suggestion! Based on the results of experiment STN, the amount of $NH_3$ (about 4.8 ppb) that contributed to $NH_4$ salt was calculated. Besides, according to the equilibrium between aerosol ($NH_4NO_3$+$(NH_4)_2SO_4$) and gas phase ($NH_3$+$HNO_3$+$H_2SO_4$), the gas phase $NH_3$ concentration was estimated to be about 3.0 ppb using the AIM Aerosol Thermodynamics Model. Therefore, the background $NH_3$ was estimated to be around 8 ppb. This information has been added in the revised manuscript.

**Revision in the manuscript:**

**Lines 334-336, Add:** "It was estimated to be around 8 ppb based on the amount of ammonium salt and the gas-aerosol equilibrium calculated using the AIM Aerosol Thermodynamics Model."

*Line 267, "in which the concentrations of NH3 were more than twice the SO2 concentrations and the oxidation products of SO2 and NOx were fully neutralized by NH3." Any experimental data or calculations to support this argument.*

**Response:** As we mentioned earlier, the concentration of $NH_3$ was not measured in this study due to the lack of appropriate instruments. The initial concentration of $NH_3$ was estimated according to the amount of $NH_3$ added into the chamber and the volume of the reactor. The chemical composition of the aerosols was measured by the AMS. The concentrations of sulfate, nitrate, and ammonium salt could be used to calculate the acid-base balance. Some data are shown in Fig. R5. This figure has been added in the supporting information. As indicated in the figure, the sulfate and nitrate was fully neutralized by ammonium. The redundant ammonium was due to the formation of organic ammonium, which was discussed in the section "Secondary organic aerosol formation" and the supporting information about estimating concentrations of nitrogen-containing organics (NOC).

[Figure]

**Figure R5.** Evaluation of acid-base balance of the aerosols according to the concentrations of sulfate, nitrate, and ammonium salt measured by the AMS in NH$_3$-rich experiments. The line in the figure indicated an acid-base balance.

**Revision in the manuscript:**

**Lines 339-344, Change:** "in which the concentrations of NH$_3$ were more than twice the SO$_2$ concentrations and the oxidation products of SO$_2$ and NOx were fully neutralized by NH$_3$."

**To:** "in which the estimated concentrations of NH$_3$ were more than twice the SO$_2$ concentrations and the oxidation products of SO$_2$ and NO$_x$ were fully neutralized by NH$_3$, according to the chemical composition of aerosols measured by the AMS. The details of the acid-base balance in the aerosols are shown in Fig. S4 in the supporting information."

**Add Fig. R5 in the supporting information**

*Line 274, "In these experiments, the concentrations of toluene were reduced compared to the experiments in Table 1, and monodisperse Al2O3 seed particles with mode diameter about 100 nm were introduced into the chamber." Any reason why a lower concentration of toluene is used for the Al2O3 seed particle experiments.*

**Response:** For experiments in Table 1, the concentrations of toluene were designed to

be high to generate enough high concentration of secondary aerosol to reduce the experimental error for a qualitative study. For experiments in Table 2, in the presence of $Al_2O_3$ seed particles, secondary aerosol would be generated on their surface and would be easier to be detected by the AMS due to a larger diameter, so lower concentrations of toluene was used to simulate secondary aerosol formation under experimental conditions closer to real ambient conditions. Some of these explanations have been added in the revised manuscript.

**Revision in the manuscript:**

**Lines 349-351, Add:** "to simulate secondary aerosol formation under experimental conditions closer to real ambient conditions"

*Line 279, "In the experiments carried out in the presence of Al2O3 seed aerosols, the decrease of NH4NO3 was not obvious in the experiment carried out in the absence of SO2 under NH3-rich conditions, indicating that generation of NH4NO3 was highly dependent on the surface area concentration of the particles, which decreased the partitioning of NH4NO3 back to the gas phase, as discussed above". In addition to describe the results in text, can the authors show the time variation of the chemical species in the SOA generated from the toluene oxidation in the presence of Al2O3 seed aerosols to support their argument?*

**Response:** Time variations of chemical species for the secondary aerosol are shown in Fig. R6. This picture has also been added in the supporting information, and corresponding description had been added in the revised manuscript. As shown in Fig. R6(b), the decrease of $NH_4NO_3$ was less than 25%, which was less obvious than experiment ATN (decrease 75%). As we mentioned earlier, based on the simulation results for the partition of $NH_4NO_3$ using the AIM Aerosol Thermodynamics Model, the concentrations of $NH_3$ gas and coexisted $(NH_4)_2SO_4$ both influenced the partition balance between $NH_4NO_3$ and $HNO_3+NH_3$ in the gas phase. The deposition of $NH_3$ gas and $(NH_4)_2SO_4$ were likely to shift balance to the gas phase and reduce the concentration of $NH_4NO_3$ salt. While the concentration of $NH_4NO_3$ salt seemed not to be affected by the deposition of $NH_4NO_3$, as long as the wall loss was corrected

accurately. We has revised the argument about the partition of NH₄NO₃ in the revised manuscript.

[Figure]

**Figure R6.** Time variations of sulfate, nitrate, and ammonium and organics measured by the AMS in the photooxidation of toluene/NO$_x$ with different concentrations of SO$_2$ under NH$_3$-poor and NH$_3$-rich conditions

**Revision in the manuscript:**

**Lines 354-362, Change:** "In the experiments carried out in the presence of Al$_2$O$_3$ seed aerosols, the decrease of NH$_4$NO$_3$ was not obvious in the experiment carried out in the absence of SO$_2$ under NH$_3$-rich conditions, indicating that generation of NH$_4$NO$_3$ was highly dependent on the surface area concentration of the particles, which decreased the partitioning of NH$_4$NO$_3$ back to the gas phase, as discussed above."

**To:** "In the experiments carried out in the presence of $Al_2O_3$ seed aerosols, the decrease of $NH_4NO_3$ was less obvious in the experiment carried out in the absence of $SO_2$ under $NH_3$-rich conditions than in experiment ATN, as indicated in Fig.S5 in the supporting information and Fig.3. This might also indicate that generation of $NH_4NO_3$ was dependent on the surface area concentration of the particles, which decreased the partitioning of $NH_4NO_3$ back to the gas phase, as discussed above concerning the effects of co-existing $(NH_4)_2SO_4$."

**Add Fig. R6 in the supporting information**

*Line 297, "A better correlation was found between secondary aerosol formation and particle surface area than that with particle volume, with details introduced in Fig. S1 in the supporting information, indicating an enhancement effect in the heterogeneous process rather than in bulk reactions". From the Figure S1, the authors argue that there is a significant correlation between the aerosol formation and aerosol surface area. From that point, they suggest that the heterogeneous processes is important than the bulk reactions. However, a strong conclusion cannot be easily drawn from the simple correlation method and limited data points. Furthermore, a nice correlation between the aerosol formation and aerosol volume is observed. Can the authors further elaborate this point?*

**Response:** We also notice that argue the heterogeneous reactions is important form the correlation is not convictive. The statement about heterogeneous and bulk reactions has been deleted in the revised manuscript. The possible importance of heterogeneous process was discussed in Page 22 based on Figure 6.

**Revision in the manuscript:**

**Lines 375-379, Delete:** "A better correlation was found between secondary aerosol formation and particle surface area than that with particle volume, with details introduced in Fig. S1 in the supporting information, indicating an enhancement effect in the heterogeneous process rather than in bulk reactions."

*Line 330, "According to the consumption of toluene, OH concentrations in the*

*photooxidation experiments were estimated to range from 1.6x106 molecules/cm3 to 2.7 x106 molecules/cm3. The reaction between these OH radicals and SO2 contributed 35%-50% of the total SO2 degradation in NH3-poor experiments, while this ratio was reduced to 25%-30% in NH3-rich experiments" Again, what is the amount of toluene and SO2 reacted in the experiments? What the time variations of the toluene and SO2 and other gas phase species (e.g. NOx) in these experiments?*

**Response:** As mentioned earlier, the time variations of toluene and $SO_2$, $O_3$ and $NO_x$ are shown in Fig. R2, while the reacted amount of toluene and $SO_2$ are shown in Table R2. This figure has been added in the supporting information. We estimated OH radical concentrations according to the decay of toluene, and then calculated the decay of $SO_2$ in the reaction between OH radical and $SO_2$ using the known reaction constant and the measured time variations of $SO_2$ concentrations, and finally calculated the ratios mentioned in the text.

**Revision in the manuscript:**

**Add Fig. R2 in the supporting information**

*Line 341, "According to previous studies, NH3 might provide surface Lewis basicity and liquid surface layers for SO2 absorption and subsequent oxidation, and therefore, enhance sulfate formation (Yang et al., 2016; Tursic et al., 2004)." What is the physical state of the aerosols in the experiments? Aqueous droplets or solid particles?*

**Response:** $Al_2O_3$ seed aerosols were used in this study. It is quite hydrophobic and would be solid in the experiments. Yang et al. reported that the coexistence of $NH_3$ would enhance the heterogeneous oxidation of $SO_2$ on $Al_2O_3$ surface(Yang et al., 2016). During the reaction, secondary aerosol would be generated on the surface of the seed aerosols or as new particles. The deliquescence relative humidity of $NH_4NO_3$ and $(NH_4)_2SO_4$ were reported to be about 60% and 80%, which were higher than the RH of 50% in the experiments. However, as we mentioned earlier, inorganic aerosol might deliquesce at a RH lower than the DRH for pure salt (Meyer et al., 2009;Li et al., 2014). Tursic et al. found the presence of $NH_3$ would significantly increase the amount of condensed water and enhance the conversion of $SO_2$ to sulfate(Tursic et al.,

2004). Thus we speculated that there might be liquid surface layers on the secondary aerosols.

**Revision in the manuscript:**

**Lines 419-424, Change:** "According to previous studies, $NH_3$ might provide surface Lewis basicity and liquid surface layers for $SO_2$ absorption and subsequent oxidation, and therefore, enhance sulfate formation (Yang et al., 2016; Tursic et al., 2004)."

**To:** "According to previous studies, $NH_3$ might provide surface Lewis basicity for $SO_2$ absorption on $Al_2O_3$ aerosols (Yang et al., 2016) and increase the amount of condensed water on the secondary aerosols (Tursic et al., 2004), and therefore, enhance sulfate formation (Yang et al., 2016;Tursic et al., 2004)"

*Line 416, "The higher production of Factor 2 with higher SO2 under an NH3-poor environment could be probably attributed to the well-known acid-catalysis effects of the oxidation product of SO2, i.e. sulfuric acid, on heterogeneous aldol condensation (Offenberg et al., 2009; Jang et al., 2002; Gao et al., 2004)" What is the pH or acidity of the aerosols in these experiments?*

**Response:** Based on the chemical compositions of aerosols measured by the AMS, the mole concentrations of species are shown in Table R3. As shown in the table, the concentration of $H^+$ increased with the increasing of $SO_2$ in $NH_3$-poor condition.

Table R3. The mole concentrations of chemical species in the $NH_3$-poor experiments in the chamber ($\mu mol/m^3$)

|  | $H^+$ | $NH_4^+$ | $SO_4^{2-}$ | $NO_3^-$ |
|---|---|---|---|---|
| 0ppb $SO_2$ | 0.000 | 0.0020 | 0.0003 | 0.0015 |
| 52ppb $SO_2$ | 0.018 | 0.016 | 0.011 | 0.011 |
| 105ppb $SO_2$ | 0.059 | 0.018 | 0.027 | 0.022 |

Using the AIM model as we mentioned earlier, the concentrations of these species in the aqueous phase was calculated. The results for the two experiments in the presence of $SO_2$ are shown in Table R4. The pH of the aerosols was calculated to be -0.5 and -0.7 in the two experiments, respectively. With this acidity, the aerosol was likely to enhance SOA formation due to acid-catalytic effects on heterogeneous aldol

condensation.

Table R4. The concentrations of inorganic species in the aqueous phase in the two $NH_3$-poor

experiments in the presence of $SO_2$

(a)   52 ppb $SO_2$

| Species | Moles | Grams | Molality | Mole Frac. | Act. Coeff. |
|---|---|---|---|---|---|
| H(aq) | 7.63E-09 | 7.69E-09 | 6.54E+00 | 6.88E-02 | 1.05E+01 |
| $NH_4$(aq) | 1.60E-08 | 2.89E-07 | 1.37E+01 | 1.44E-01 | 2.63E-01 |
| $HSO_4$(aq) | 1.04E-08 | 1.01E-06 | 8.89E+00 | 9.35E-02 | 3.77E+00 |
| $SO_4$(aq) | 1.13E-09 | 1.08E-07 | 9.67E-01 | 1.02E-02 | 7.80E-03 |
| $NO_3$(aq) | 1.10E-08 | 6.82E-07 | 9.43E+00 | 9.92E-02 | 3.84E-01 |
| OH(aq) | 1.68E-26 | 2.85E-25 | 1.44E-17 | 1.51E-19 | 2.13E+01 |
| $H_2O$(aq) | 6.48E-08 | 1.17E-06 | 5.55E+01 | 5.84E-01 | 8.56E-01 |
| $NH_3$(aq) | 4.90E-20 | 8.34E-19 | 4.20E-11 | 4.42E-13 | 1.71E+00 |

The density of the aqueous phase is 1.39289 g per $cm^3$, and its total volume is 2.34095E-06 $cm^3$.

(b)   105 ppb $SO_2$

| Species | Moles | Grams | Molality | Mole Frac. | Act. Coeff. |
|---|---|---|---|---|---|
| H(aq) | 3.50E-08 | 3.53E-08 | 8.93E+00 | 1.09E-01 | 8.35E+00 |
| $NH_4$(aq) | 1.80E-08 | 3.25E-07 | 4.59E+00 | 5.62E-02 | 1.85E-01 |
| $HSO_4$(aq) | 2.40E-08 | 2.33E-06 | 6.12E+00 | 7.49E-02 | 1.00E+01 |
| $SO_4$(aq) | 3.51E-09 | 3.37E-07 | 8.95E-01 | 1.10E-02 | 1.22E-02 |
| $NO_3$(aq) | 2.20E-08 | 1.36E-06 | 5.61E+00 | 6.87E-02 | 8.43E-01 |
| OH(aq) | 5.31E-25 | 9.03E-24 | 1.35E-16 | 1.66E-18 | 1.54E+00 |
| $H_2O$(aq) | 2.18E-07 | 3.92E-06 | 5.55E+01 | 6.80E-01 | 7.35E-01 |
| $NH_3$(aq) | 3.57E-20 | 6.07E-19 | 9.09E-12 | 1.11E-13 | 1.47E+00 |

The density of the aqueous phase is 1.34747 g per $cm^3$, and its total volume is 6.16843E-06 $cm^3$.

**Revision in the manuscript:**

**Lines 505-508, Add:** "This is consistent with the fact that the aerosols in the

NH₃-poor environment were quite acidic according to the simulation results of the AIM model, based on the chemical compositions of aerosols measured by the AMS."

*Line 427 "By inference and from the results of AMS measurements, aerosol water increased as the initial concentration of SO2 increased, since more inorganic aerosol was generated. Liggio and Li (2013) suggest that dissolution of primary polar gases into a partially aqueous aerosol contributed to the increase of organic mass and oxygen content on neutral and near-neutral seed aerosols, which would also take place in the NH3-rich experiments and contribute to the generation of Factor 1."*
*What is the amount of aerosol phase water and how the aerosol water content change with reactions in these experiments?*

**Response:** The aerosol water concentrations measured by AMS are shown in Figure R7. First, we have to point out that there is high uncertainty about the aerosol water measured by AMS. But there is a tendency than aerosol water increased with reaction time. Besides, higher concentrations of aerosol water were observed in NH₃-rich experiment compare to that in the NH₃-poor experiment.

[Figure]

**Figure R7.** Time variations of aerosol water measured by the AMS in the photooxidation of

**References**

Anttila, T., Kiendler-Scharr, A., Tillmann, R., and Mentel, T. F.: On the reactive uptake of gaseous compounds by organic-coated aqueous aerosols: Theoretical analysis and application to the heterogeneous hydrolysis of N$_2$O$_5$, J. Phys. Chem. A, 110, 10435-10443, doi: 10.1021/jp062403c, 2006.

Atkinson, R., Baulch, D. L., Cox, R. A., Crowley, J. N., Hampson, R. F., Hynes, R. G., Jenkin, M. E., Rossi, M. J., and Troe, J.: Evaluated kinetic and photochemical data for atmospheric chemistry: Volume I - gas phase reactions of Ox, HOx, NOx and SOx species, Atmos. Chem. Phys., 4, 1461-1738, doi: doi:10.5194/acp-4-1461-2004, 2004.

Carslaw, K. S., Clegg, S. L., and Brimblecombe, P.: A thermodynamic model of the system HCl-HNO$_3$-H$_2$SO$_4$-H$_2$O, including solubilities of HBr, from less-than-200 to 328 K, J. Phys. Chem., 99, 11557-11574, doi:10.1021/j100029a039, 1995.

Chu, B., Hao, J., Takekawa, H., Li, J., Wang, K., and Jiang, J.: The remarkable effect of FeSO$_4$ seed aerosols on secondary organic aerosol formation from photooxidation of α-pinene/NOx and toluene/NOx, Atmos. Environ., 55, 26-34, doi:10.1016/j.atmosenv.2012.03.006, 2012.

Chu, B., Liu, Y., Li, J., Takekawa, H., Liggio, J., Li, S.-M., Jiang, J., Hao, J., and He, H.: Decreasing effect and mechanism of FeSO$_4$ seed particles on secondary organic aerosol in α-pinene photooxidation, Environ. Pollut., 193, 88-93, doi:10.1016/j.envpol.2014.06.018, 2014.

Clegg, S. L., Brimblecombe, P., and Wexler, A. S.: Thermodynamic model of the system H$^+$-NH$_4^+$-SO$_4^{2-}$-NO$_3^-$-H$_2$O at tropospheric temperatures, J. Phys. Chem. A, 102, 2137-2154, doi:10.1021/jp973042r, 1998.

Clegg, S. L., and Brimblecombe, P.: Comment on the "Thermodynamic dissociation constant of the bisulfate ion from Raman and ion interaction modeling studies of aqueous sulfuric acid at low temperatures", J. Phys. Chem. A, 109, 2703-2706, doi:10.1021/jp0401170, 2005.

Hallquist, M., Stewart, D. J., Stephenson, S. K., and Cox, R. A.: Hydrolysis of N2O5 on sub-micron sulfate aerosols, Phys. Chem. Chem. Phys., 5, 3453-3463, doi: 10.1039/b301827j, 2003.

Hildebrandt Ruiz, L., Paciga, A. L., Cerully, K. M., Nenes, A., Donahue, N. M., and Pandis, S. N.: Formation and aging of secondary organic aerosol from toluene: changes in chemical composition, volatility, and hygroscopicity, Atmos. Chem. Phys., 15, 8301-8313, doi:10.5194/acp-15-8301-2015, 2015.

Hu, J. H., and Abbatt, J. P. D.: Reaction probabilities for N$_2$O$_5$ hydrolysis on sulfuric acid and ammonium sulfate aerosols at room temperature, J. Phys. Chem. A, 101, 871-878, doi: 10.1021/jp9627436, 1997.

Li, W. J., Shao, L. Y., Shi, Z. B., Chen, J. M., Yang, L. X., Yuan, Q., Yan, C., Zhang, X. Y., Wang, Y. Q., Sun, J. Y., Zhang, Y. M., Shen, X. J., Wang, Z. F., and Wang, W. X.: Mixing state and hygroscopicity of dust and haze particles before leaving Asian continent, J. Geophys. Res.- Atmos., 119, 1044-1059, doi:10.1002/2013jd021003,

2014.

Meyer, N. K., Duplissy, J., Gysel, M., Metzger, A., Dommen, J., Weingartner, E., Alfarra, M. R., Prevot, A. S. H., Fletcher, C., Good, N., McFiggans, G., Jonsson, A. M., Hallquist, M., Baltensperger, U., and Ristovski, Z. D.: Analysis of the hygroscopic and volatile properties of ammonium sulphate seeded and unseeded SOA particles, Atmos. Chem. Phys., 9, 721-732, doi: 10.5194/acp-9-721-2009, 2009.

Odum, J. R., Jungkamp, T. P. W., Griffin, R. J., Forstner, H. J. L., Flagan, R. C., and Seinfeld, J. H.: Aromatics, reformulated gasoline, and atmospheric organic aerosol formation, Environ. Sci. & Technol., 31, 1890-1897, doi:10.1021/es960535l, 1997.

Sato, K., Hatakeyama, S., and Imamura, T.: Secondary organic aerosol formation during the photooxidation of toluene: NOx dependence of chemical composition, J. Phys. Chem. A, 111, 9796-9808, doi:10.1021/jp071419f, 2007.

Takekawa, H., Minoura, H., and Yamazaki, S.: Temperature dependence of secondary organic aerosol formation by photo-oxidation of hydrocarbons, Atmos. Environ., 37, 3413-3424, doi:10.1016/s1352-2310(03)00359-5, 2003.

Tursic, J., Berner, A., Podkrajsek, B., and Grgic, I.: Influence of ammonia on sulfate formation under haze conditions, Atmos. Environ., 38, 2789-2795, doi:10.1016/j.atmosenv.2004.02.036, 2004.

Yang, W., He, H., Ma, Q., Ma, J., Liu, Y., Liu, P., and Mu, Y.: Synergistic formation of sulfate and ammonium resulting from reaction between $SO_2$ and $NH_3$ on typical mineral dust, Phys. Chem. Chem. Phys., 18, 956-964, doi:10.1039/c5cp06144j, 2016.

---

## Author Comment (AC2) · 31 Oct 2016

**Response to Reviewer #2**

**Ms. Ref. No.: acp-2016-486**

**Title: "Synergetic formation of secondary inorganic and organic aerosol: Influence of SO₂ and/or NH₃ in the heterogeneous process"**

**Revised Title: "Synergetic formation of secondary inorganic and organic aerosol: Effect of SO₂ and NH₃ on particle formation and growth"**

We appreciate the comments from the reviewer on this manuscript. We have answered them in the following paragraphs (the text in italics is the reviewer comments, followed by our response) point by point. The text in blue is some important revisions for the manuscript. The line numbers in the response are from the revised manuscript.

*The manuscript reports data related to the effect of SO2 and NH3 on aerosol formation from the oxidation of toluene in the presence on NOx. The experimental study was conducted in the presence or absence of inorganic seed aerosol: AL2O3. NH3 and SO2 are two species emitted into the atmosphere and can have a large effect on atmospheric chemistry. The data analysis show aerosol formation and growth increased in the presence of SO2 regardless of the presence of NH3. This study and its topic is of great interest and appropriate to ACPD journal. This study is worth to be published since it present an important set of data that can be useful for the atmospheric communities. However, I feel that the text and the scientific discussion (interpretation of experimental data) (see my comments below) need to be addressed before publication.*

*The experimental part needs to be addressed and clearly state how the experiments were run. Toluene as well other gas phase species need to be reported vs time in this study. The role of OH radicals should be discussed? The wall loss of gas phase and particles should be addressed also? The errors and uncertainties need to be*

*addressed since assumptions were made in this study. Yield should be reported in this study for the different systems studied.*

**Response:** Thanks for the reviewer's comments. Here we response to some of the comments, while the other questions (including gas compounds, OH radicals, wall loss and yield) would be answered later in the following paragraphs point by point.

Some additional information about how we run the experiments has been added in the methods section. We introduced how we wash the chamber, how we control the humidity, how we adding gas and particles into the chamber and so on.

Several assumptions were made in this study due to the limitation of analytical instruments. In Line 184 in the original manuscript, we assumed a same aerosol density in experiments in the presence or absence of $NH_3$ or $SO_2$. This assumption is actually not true. In the presence of $NH_3/SO_2$, more inorganic aerosol (higher mass proportion) was generated than the experiment in the absence of $NH_3/SO_2$. The density of inorganic aerosol, mainly sulfate, nitrate and ammonium, is about 1.7 $g/cm^3$, while it is about 1.4 $g/cm^3$ for SOA. Therefore, the assumption of a same aerosol density would under estimate the increase effect of $NH_3$ or $SO_2$ on secondary aerosol formation. To keep things simple and to avoid misunderstanding, this sentence has been deleted in the revised manuscript. In Line 184 in the original manuscript, we assumed that the presence of $SO_2$ and $NH_3$ did not significantly impact the gas phase oxidation of hydrocarbons and mainly played a role in the aerosol phase. This is actually not an assumption but a corollary based on previous studies and experimental data in this study. Some revision about these sentences has been made in the revised manuscript. The assumption and uncertainties about $NH_3$ concentrations would be discussed later in this file.

**Revision in the manuscript:**

For how the experiments were run:

**Lines 179-183, Add:** "Prior to each experiment, the chamber was flushed for about 40 h with purified air at a flow rate of 15 L/min. In the first 20 h, the chamber was

exposed to UV light at 34 ℃. In the last several hours of the flush, humid air was introduced to obtain the target RH, which was 50% in this study. After that, alumina seed particles were added into the chamber."

**Lines 194-204, Add:** "$NO_x$, $SO_2$ and $NH_3$ were directly injected into the chamber from standard gas bottles using mass flow controllers. Before adding $NH_3$ into the chamber, $NH_3$ gas was passed through the inlet pipeline for about 15 minutes to reduce absorption within the line. The concentrations of $NH_3$ were estimated according to the amount of $NH_3$ introduced and the volume of the reactor. These experiments with $NH_3$ added to the chamber were referred to as $NH_3$-rich experiments in this study, since the concentrations of $NH_3$ were not measured and it was difficult to estimate the uncertainty of the calculated $NH_3$ concentration."

**Lines 205-209, Add:** "The experiments were carried out at 30 ℃ with an initial RH of 50%. During the reaction, the temperature was kept nearly constant (30±0.5 ℃) in the temperature-controlled enclosure, while the RH decreased to 45%-47% at the end of the experiment."

For the errors and uncertainties:

**Lines 192-194, Delete:** "Assuming the same aerosol density in these experiments, the presence of either $NH_3$ or $SO_2$ enhanced secondary aerosol formation markedly."

**Line 426, Change:** "assumed"

**To:** "speculated"

*As I mentioned, the manuscript reports a set of great data important to scientist interested in atmospheric organic and inorganic aerosol formation and the effect of NH3 and SO2!*

**Response:** Thanks for the affirmation.

*Comments:*

*In the introduction (1st paragraph), the authors report literature data for NH3 in China and almost no data was provided for SO2. I suggest SO2 should be provided also and a comparison should be reported between SO2 and NH3. The text in the*

*manuscript should be edited for consistency. I found it very hard to follow the authors'*
*ideas in the manuscript, although lot of information is provided. For example,*
*sentences reported between lines 89 and 115 are very difficult to follow for me!!! This*
*is true for most the manuscript!*

**Response:** We agree with the reviewer that more literature data for $SO_2$ are needed. More information about $SO_2$ and a simple comparison between $SO_2$ and $NH_3$ has also been added in the introduction.

The introduction, the description about the results and the discussions were carefully amended in the revised manuscript. The manuscript was also revised by a native English to make it more readable. A lot of revisions have been made. These revisions are recorded in the revised manuscript, but are not listed one by one here.

**Revision in the manuscript:**

**Lines 61-72, Change:** "For example, the $SO_2$ concentration in Jinan, a city in North China, can be as high as 43 ppb in the winter season (Wang et al., 2015a)"

**To:** "China has the highest concentration of $SO_2$ in the world due to a large proportion of energy supply from coal combustion (Bauduin et al., 2016). Surface concentrations of $SO_2$ in the range of a few ppb to over 100 ppb have been observed in north China (Sun et al., 2009; Li et al., 2007). The total emission and concentrations of $SO_2$ have decreased in most regions of China in recent years (Lu et al., 2010; Wang et al., 2015b), but high concentrations of $SO_2$ are still frequently observed. For example, the $SO_2$ concentration was as high as 43 ppb in the winter of 2013 in Jinan city (Wang et al., 2015a), while over 100 ppb $SO_2$ was observed in winter haze days during 2012 in Xi'an city (Zhang et al., 2015)."

**Lines 83-85, Add:** "Unlike $SO_2$, the emission of $NH_3$ is mainly from non-point sources, which are difficult to control, and shows an increasing trend in China (Dong, 2010)."

*Line 125: this study focusses on "toluene" and "VOC" should be deleted.*

**Response:** Corresponding revision has been made in the revised manuscript.

**Revision in the manuscript:**

**Line 141, Change:** "VOC/NO$_x$"

**To:** "toluene/NO$_x$"

*The chamber was a 2 m3 and the losses expected to be higher. The authors should give more information in this study about the wall losses of gas phase and particles.*

**Response:** Yes, the wall deposition of particles was high in this study due to the small volume of the chamber. We measured the deposition rate of different gases, particles of different sizes. Deposition of particles and gas compounds on the wall was considered to be a first-order process for wall loss correction. Additional information has been added in the revised manuscript.

**Revision in the manuscript:**

**Lines 165-169, Add:** "The chamber was run as a batch reactor in this study. Deposition of particles and gas compounds on the wall was considered to be a first-order process. The deposition rates of particles with different sizes (40-700 nm) were measured under dark conditions."

**Lines 173-178, Add:** "The deposition of gas phase compounds was determined to be 0.0025 h$^{-1}$, 0.0109 h$^{-1}$, 0.0023 h$^{-1}$ and 0.006 h$^{-1}$ for NO$_2$, O$_3$, NO and toluene, respectively. In this study, the wall loss of aerosol mass was about 30%-50% of total secondary aerosol mass, while the deposition of gas phase compounds was less than 5% of their maximum concentrations in the experiments."

*How NH3 was estimated (line 168)? Needs errors and uncertainties?*

**Response:** The concentration of NH$_3$ is an important problem in this study. Due to the lack of analytical instruments, we were not able to measure the concentrations of NH$_3$ in the chamber. Alternatively, we used a mass flow controller and a stopwatch to control the added amount of NH$_3$ in the chamber. Then, we estimated the initial concentrations of NH$_3$ based on the volume of the reactor. As the reviewer pointed out, there are many uncertainties for the estimated concentrations. The main uncertainties included:

(1) the absorption of NH$_3$ in the inlet pipeline (Before adding NH$_3$ into the chamber,

NH$_3$ gas was passed through the inlet pipeline for about 15 minutes to reduce absorption within the line. This uncertainty is difficult to estimate.)

(2) the concentrations of NH$_3$ standard gas (Less than 1%, Beijing AP BAIF Gases Industry CO., Ltd)

(3) the volume of the reactor (Less than 5% according to our experiences and the concentrations of other pollutants)

(4) the background NH$_3$ gas in the chamber (As we mentioned in the manuscript, there was NH$_3$ present in the background air in the chamber derived from the partitioning of the deposited ammonium sulfate and nitrate on the chamber wall when humid air was introduced. Based on the results of experiment STN (without NH$_3$ added), the amount of NH$_3$ that contributed to NH$_4$ salt was calculated to be about 4.8 ppb. Besides, according to the equilibrium between aerosol (NH$_4$NO$_3$+(NH$_4$)$_2$SO$_4$) and gas phase (NH$_3$+HNO$_3$+H$_2$SO$_4$), the gas phase NH$_3$ concentration was estimated to be about 3.0 ppb using the AIM Aerosol Thermodynamics Model. The detail of the model is available at http://www.aim.env.uea.ac.uk/aim/aim.php, and was described elsewhere (Clegg and Brimblecombe, 2005;Clegg et al., 1998;Carslaw et al., 1995). Therefore, the background NH$_3$ was estimated to be around 8 ppb. This information has been added in the revised manuscript.)

**Revision in the manuscript:**

**Lines 196-204, Add:** "NO$_x$, SO$_2$ and NH$_3$ were directly injected into the chamber from standard gas bottles with mass flow controllers. Before adding NH$_3$ into the chamber, NH$_3$ gas was passed the inlet pipeline for about 15 minutes to reduce the absorption. The concentrations of NH$_3$ were then estimated according to the introduced amount of NH$_3$ and the volume of the reactor. These experiments with NH$_3$ added to the chamber were referred as NH$_3$-rich experiments in this study since the concentrations of NH$_3$ were not measured and it was difficult to estimate the uncertainty of the calculated NH$_3$ concentration."

**Lines 333-335, Add:** "It was estimated to be around 8 ppb based on the amount of ammonium salt and the gas-aerosol equilibrium calculated using the AIM Aerosol Thermodynamics Model. "

*Line 171. title not clear to me? Be specific I would suggest: Effect of NH3 and SO2 on particle formation and growth*

**Response:** Corresponding revision has been made in the revised manuscript.

**Revision in the manuscript:**

**The title, Change:** "Synergetic formation of secondary inorganic and organic aerosol: Influence of $SO_2$ and/or $NH_3$ in the heterogeneous process"

**To:** "Synergetic formation of secondary inorganic and organic aerosol: Effect of $SO_2$ and $NH_3$ on particle formation and growth"

*Table 1 change "hydrocarbon" to "toluene"*

**Response:** Corresponding revision has been made in the revised manuscript.

**Revision in the manuscript:**

**Table 1, Change:** "hydrocarbon"

**To:** "toluene"

*Data provided in Table 1 are initial concentration? The authors need to specify how the chamber was run (as flow reactor or batch reactor). Based on fig 1, it seems to me it was conducted as a batch reactor. In this case the wall losses are important and need to be incorporated in the discussion and how toluene and SMPS data were analyzed to get to Figs 1, 2.. and tables 1, 2.*

**Response:** Yes, the data provided in Table 1 are the initial concentrations. The chamber was run as a batch reactor. To make this clear, corresponding revisions have been made in the revised manuscript. For the wall losses, as we mentioned earlier, additional information about the deposition rates of gas phase compounds and contribution of particle deposition to Figs 1 and 2 have been added in the revised manuscript.

**Revision in the manuscript:**

**Table 2, Change:** "Experimental conditions"

**To:** "Initial experimental conditions"

(For Table 1, it is "Initial experimental conditions" in the manuscript.)

**Lines 165-169 Add:** "The chamber was run as a batch reactor in this study. Deposition of particles and gas compounds on the wall was considered to be a first-order process. The deposition rates of particles with different sizes (40-700 nm) were measured under dark conditions."

**Lines 173-178, Add:** "The deposition of gas phase compounds was determined to be 0.0025 $h^{-1}$, 0.0109 $h^{-1}$, 0.0023 $h^{-1}$ and 0.006 $h^{-1}$ for $NO_2$, $O_3$, NO and toluene, respectively. In this study, the wall loss of aerosol mass was about 30%-50% of total secondary aerosol mass, while the deposition of gas phase compounds was less than 5% of their maximum concentrations in the experiments."

*It is also very important to provide data for Toluene reacted, SO2 reacted, NH3 reacted and NO reacted, NOx reacted (vs. time) in the gas phase in a separate figure.*

**Response:** Time variations of gas-phase compounds in photooxidation of toluene/$NO_x$ in the presence or absence of $NH_3$ and/or $SO_2$ are displayed in Figure R1 and Figure R2. These Figures has been added in the revised Supporting information.

[Figure]

**Figure R1.** Time variations of gas-phase compounds in photooxidation of toluene/NO$_x$ in the presence or absence of NH$_3$ and/or SO$_2$. The letters codes for the experiments indicate the introduced pollutants, i.e. "A" for ammonia, "S" for sulfur dioxide, "T" for toluene and "N" for nitrogen dioxide.

[Figure]

**Figure R2.** Time variations of gas-phase compounds in photooxidation of toluene/NO$_x$ with different concentrations of SO$_2$ under NH$_3$-poor and NH$_3$-rich conditions

**Revision in the manuscript:**

**Lines 222-225, Add:** "Time variations of gas phase compounds of these experiments are shown in Fig. S1 in the supporting information. The presence of SO$_2$ and/or NH$_3$ had no obvious effect on the gas phase compounds, including toluene, NO$_x$, SO$_2$ and O$_3$."

**Add Fig. R1 and Fig. R2 in the supporting information**

*The experiments were conducted under 50% RH. The authors didn't reported in the text until Table 1 was mentioned. This is a very important parameters that should be reported and discussed at list briefly. How it was measured and controlled in the*

*chamber!!! Same thing for the temperature! I'm expecting the RH will change aver the run time and will be not constant?*

**Response:** Thanks for the reviewer's reminding. As we mentioned above for how the experiment was run, some information about the temperature and RH have been added in the methods in the revised manuscript.

**Revision in the manuscript:**

**Lines 179-183, Add:** "Prior to each experiment, the chamber was flushed for about 40 h with purified air at a flow rate of 15 L/min. In the first 20 h, the chamber was exposed to UV light at 34 ℃. In the last several hours of the flush, humid air was introduced to obtain the target RH, which was 50% in this study. After that, alumina seed particles were added into the chamber."

**Lines 205-209, Add:** "The experiments were carried out at 30 ℃ with an initial RH of 50%. During the reaction, the temperature was kept nearly constant (30±0.5 ℃) in the temperature-controlled enclosure, while the RH decreased to 45%-47% at the end of the experiment."

*How NH3 was introduced into the chamber? Please elaborate!*

**Response:** $NH_3$ was introduced into the chamber using standard gas from high-pressure gas bottle. We used a mass flow controller, a stopwatch and a solenoid three-way valve to control the amount of $NH_3$ in the chamber. Then, we estimated the initial concentrations of $NH_3$ based on the volume of the reactor. The solenoid three-way valve was placed on the inlet next to the chamber. Before adding $NH_3$ into the chamber, $NH_3$ gas was passed the inlet pipeline for about 15 minutes (not into the chamber through the valve) to reduce the absorption when adding $NH_3$ into the chamber.

**Revision in the manuscript:**

**Lines 196-204, Add:** "$NO_x$, $SO_2$ and $NH_3$ were directly injected into the chamber from standard gas bottles using mass flow controllers. Before adding $NH_3$ into the chamber, $NH_3$ gas was passed through the inlet pipeline for about 15 minutes to reduce absorption within the line. The concentrations of $NH_3$ were estimated

according to the amount of $NH_3$ introduced and the volume of the reactor. These experiments with $NH_3$ added to the chamber were referred to as $NH_3$-rich experiments in this study, since the concentrations of $NH_3$ were not measured and it was difficult to estimate the uncertainty of the calculated $NH_3$ concentration."

*It's important to have the amount of toluene reacted in each case in order to see how much was oxidized. then measure the yield etc... The OH radicals present in the system can be reacted with different gas phase species (e.g. toluene, SO2,..) and then depending on the rate constant, it can affect the conclusion reported in this study.*

**Response:** The reacted amounts of gas phase species were calculated and are displayed in Table R1 and Table R2. SOA yields were also calculated. The presence of $SO_2$/$NH_3$ had no obvious effect on the reacted amount of toluene. In Fig. R3, the SOA yields in this study are compared with literature results. SOA yields in in photooxidation of toluene/$NO_x$ in the presence or absence of $NH_3$ and/or $SO_2$ were similar as that reported by Odum, J. R., et. al.. A closer inspection revealed that experiment TN had a SOA yield a little lower than the curve in the study of Odum, J. R., et. al., experiment STN and ATN had SOA yields quite close to the curve, while experiment ASTN had a yield a little higher than the curve. For SOA yields in photooxidation of toluene/$NO_x$ with different concentrations of $SO_2$, SOA yields were higher in $NH_3$-rich condition compare to $NH_3$-poor condition. And there is a trend that SOA yield increased with increasing $SO_2$ concentrations. The presence of $SO_2$/$NH_3$ increased SOA yield.

**Table R1.** The consumption of gas precursors, the formation of ozone and SOA, and SOA yield in photooxidation of toluene/$NO_x$ in the presence or absence of $NH_3$ and/or $SO_2$. The letters codes for the experiments indicate the introduced pollutants, i.e. "A" for ammonia, "S" for sulfur dioxide, "T" for toluene and "N" for nitrogen dioxide.

| Experiment No. | $\Delta$toluene ppm | $\Delta NO_x$ ppb | $\Delta SO_2$ ppb | $\Delta O_3$ ppb | SOA $\mu g/m^3$ | SOA yield % |
|---|---|---|---|---|---|---|
| TN | 0.19 | 89 | NA | 179 | 8.0 | 1.1 |
| STN | 0.18 | 90 | 25 | 176 | 27.7 | 4.1 |

| | | | | | | |
|---|---|---|---|---|---|---|
| ATN | 0.16 | 82 | NA | 159 | 17.2 | 2.9 |
| ASTN | 0.15 | 90 | 42 | 166 | 37.4 | 6.8 |

**Table R2.** The consumption of gas precursors, the formation of ozone and SOA, and SOA yield in photooxidation of toluene/$NO_x$ with different concentrations of $SO_2$ under $NH_3$-poor and $NH_3$-rich conditions

| | Δtoluene
ppb | Δ$NO_x$
ppb | Δ$SO_2$
ppb | Δ$O_3$
ppb | SOA
μg/m$^3$ | SOA yield
% |
|---|---|---|---|---|---|---|
| | 41 | 73 | NA | 22 | 0.6 | 0.4 |
| $NH_3$-poor | 40 | 61 | 6 | 22 | 1.4 | 1.0 |
| | 39 | 62 | 12 | 20 | 1.5 | 1.1 |
| | 47 | 79 | NA | 27 | 3.5 | 2.0 |
| $NH_3$-rich | 45 | 81 | 6 | 26 | 5.1 | 3.1 |
| | 44 | 75 | 11 | 27 | 6.7 | 4.1 |

[Figure]

**Figure R3.** SOA yields in the experiments in this study and the comparison with literature results (Takekawa et al., 2003;Hildebrandt Ruiz et al., 2015;Odum et al., 1997;Sato et al., 2007)

**Revision in the manuscript:**

**Add Table. R1 and Table. R2 in the supporting information**

**Add Fig. R3 in the supporting information**

**Lines 428-430, Add:** "The increases of SOA mass in the presence of NH$_3$ and SO$_2$ are shown in Fig. 5. Similar trends for SOA yields can be found in the supporting information."

*Line 225: How the authors distinguish between secondary organic aerosol and secondary inorganic aerosol in these experiments?*

**Response:** The chemical composition of the aerosols was measured by the ACSM and AMS in this study. The measurement results of ACSM and AMS including the concentrations of sulfate, nitrate, ammonium salt and organics. In the ACSM or AMS, the aerosols were heated to about 600℃ and ionized by 70eV electrons. During the ionization, most secondary species were fragmented. There were some organic nitrates and organic acid ammonium in the aerosols, as we discussed in the section of "secondary organic aerosol formation", but they can't be identified from the ACSM or AMS data. Therefore, the measured sulfate, nitrate, ammonium salt (estimated from corresponding fragments) were all considered secondary inorganic aerosol, and the organics (estimated from organic fragments) were all considered secondary organic aerosol. Some explanation has been added in the revised manuscript.

**Revision in the manuscript:**

**Lines 280-284, Add:** "Since the ACM or AMS cannot distinguish organic salts and organic nitrates, the measured sulfate, nitrate, ammonium were all considered secondary inorganic aerosol, while the organics were all considered secondary organic aerosol in this study."

*Line 228-230 How the authors come to this conclusion? It's speculation and not based on data reported here. How nitrate are measured in this study? Are you refering to inorganic or organic nitrates? It will be great if a distinction was made between SOA and secondary inorganic aerosol in this study?*

**Response:** As we mentioned above, the chemical composition of the aerosols was measured by the ACSM and AMS in this study. The inorganic or organic nitrates

were not distinguished. Here, the nitrate is the sum of inorganic or organic nitrates. Therefore, besides ammonia nitrate, organic nitrate might also contribute to the observed nitrate.

**Revision in the manuscript:**

**Lines 276-279, Change:** "For example, nitrate formation was not only enhanced by NH₃, due to conversion of nitric acid into ammonia nitrate, but also was markedly affected by $SO_2$."

**To:** "For example, nitrate formation (which may include both inorganic nitrate and organic nitrates) was not only enhanced by $NH_3$, but also was markedly affected by $SO_2$."

*It will be interesting to estimate the concentration of OH radicals vs time in these experiments?*

**Response:** The time variations of OH radicals were calculated based on the time variations of toluene and the reaction rate between toluene and OH radical. The results are shown in the following two pictures. The concentrations are quite scattered due to the measurement error of the toluene in the GC-FID. As we can see in the two pictures, the estimated OH radical concentrations were higher at the beginning of the experiment than that at the end of the experiment.

[Figure]

**Figure R4.** Time variations of OH radical concentrations in photooxidation of toluene/NO$_x$ in the

presence or absence of NH$_3$ and/or SO$_2$. The letters codes for the experiments indicate the introduced pollutants, i.e. "A" for ammonia, "S" for sulfur dioxide, "T" for toluene and "N" for nitrogen dioxide.

[Figure]

**Figure R5.** Time variations of OH radical concentrations in photooxidation of toluene/NO$_x$ with different concentrations of SO$_2$ under NH$_3$-poor and NH$_3$-rich conditions

*Line 239. "The larger diameter resulted in more significant wall deposition, reduced the surface area of the suspended particles, and shifted the partition equilibrium to the gas phase." Are the authors measured the wall losses at different size distribution or this only speculation?*

**Response:** As we mentioned above, the deposition rates of particles with different sizes (40-700nm) under dark conditions were measured. Additional information has been added in the revised manuscript. However, the reason for the decreasing of NH$_4$NO$_3$ was revised according to the simulation results from the AIM Aerosol Thermodynamics Model. The simulation results are summarized in Fig. R6. This figure has been added in the supporting information. The results showed that the concentrations of NH$_3$ gas and coexisted (NH$_4$)$_2$SO$_4$ both influenced the partition balance between NH$_4$NO$_3$ and HNO$_3$+NH$_3$ in the gas phase. The deposition of NH$_3$ gas and (NH$_4$)$_2$SO$_4$ were likely to shift balance to the gas phase and reduce the concentration of NH$_4$NO$_3$ salt. While the concentration of NH$_4$NO$_3$ salt seemed not to be affected by the deposition of NH$_4$NO$_3$, as long as the deposition was corrected

accurately. According to these results, some revision has been made in the revised manuscript.

[Figure]

**Figure R6.** Concentrations of NH₄NO₃ salt as a function of the wall deposition of NH₃, (NH₄)₂SO₄ and NH₄NO₃ based on the simulation results of the AIM Aerosol Thermodynamics Model

**Revision in the manuscript:**

**Lines 165-169, Add:** "The chamber was run as a batch reactor in this study. Deposition of particles and gas compounds on the wall was considered to be a first-order process. The deposition rates of particles with different sizes (40-700 nm) were measured under dark conditions."

**Lines 290-302, Change:** "In 错误!未找到引用源。, we observed that the particle size was larger in experiment ATN than the other three experiments. The larger diameter resulted in more significant wall deposition, reduced the surface area of the suspended particles, and shifted the partition equilibrium to the gas phase."

**To:** "Detailed simulation results based on the AIM Aerosol Thermodynamics Model (Clegg and Brimblecombe, 2005;Clegg et al., 1998;Carslaw et al., 1995) are shown in Fig. S3 in the supporting information. The deposition of $NH_3$ in the experiment was likely to shift the partition equilibrium to the gas phase and reduce the concentration of $NH_4NO_3$ salt. In addition, the wall deposition of aerosols might also introduce some error in the concentrations of $NH_4NO_3$ salt, although wall deposition was corrected using an empirical function based on deposition rates of $(NH_4)_2SO_4$ aerosol with different sizes (Chu et al., 2012;Chu et al., 2014)."

**Add Fig. R6 in the supporting information**

*Line 245 – 249. It seems to me that these were not based on data reported in these study. How N2O5 plays a role here? N2O5 was measured in this study? How it was formed in the chamber. Is ozone was measured in this study? If yes it should be reported vs time.*

**Response:** In the presence of organic compounds, $(NH_4)_2SO_4$ was reported to deliquesce at RH lower than pure $(NH_4)_2SO_4$. (Meyer et al., 2009;Li et al., 2014). To avoid misunderstanding, the description was revised.

$N_2O_5$ was not measured in this study, but it was expected to be generated in the presence of $NO_2$ and $O_3$ in the experiments. Under the experimental conditions, the maximum formation velocity was calculated to be about 13 ppb/hour from gas phase reaction between $NO_2$ and $O_3$. The concentrations of $NO_2$ and $O_3$ are shown in Fig.

R1. The reaction constant was summarized by Atkinson et al. (2004). The uptake coefficient of $N_2O_5$ on particle surface was reported to be about $10^{-2}$ on ammonium sulfate (Hallquist et al., 2003;Hu and Abbatt, 1997), but would decrease when organics coated on the sulfate (Anttila et al., 2006). The particle surface area concentration in experiment ASTN ranged from 0 to 0.1 $m^2/m^3$. Assuming a concentration of $N_2O_5$ of 0.1 ppb and an uptake coefficient of $10^{-3}$ for $N_2O_5$, the heterogeneous hydrolysis of $N_2O_5$ on 0.05 $m^2/m^3$ suspended particles would generate 6 $\mu g/m^3$ nitrate per hour in the reactor. Thus we speculated the heterogeneous hydrolysis of $N_2O_5$ might be important in the experiment. Some of these explanations have been added in the revised manuscript.

**Revision in the manuscript:**

**Lines 308-317, Change:** "In addition, the presence of organic matter might accelerate the deliquescence of generated inorganic particles (Meyer et al., 2009;Li et al., 2014), and provide moist surfaces for heterogeneous hydrolysis of $N_2O_5$, contributing to nitrate formation (Pathak et al., 2009)."

**To:** "In addition, in the presence of organic matter, $(NH_4)_2SO_4$ aerosol might deliquesce at a RH lower than the deliquescence relative humidity (DRH) (Meyer et al., 2009;Li et al., 2014). If this took place in the experiment, sulfate might provide moist surfaces for heterogeneous hydrolysis of $N_2O_5$, contributing to nitrate formation due to the high uptake coefficient of $N_2O_5$ on ammonium sulfate (Pathak et al., 2009;Hallquist et al., 2003;Hu and Abbatt, 1997). $N_2O_5$ was not measured in this study, but it was expected to be generated in the presence of $NO_2$ and $O_3$ in the experiments."

*Lines 256. Is ammonia salt was measured? How the authors come to the measured ammonia salt.*

**Response:** As we mentioned earlier, the chemical composition of the aerosols were measured by the ACSM or AMS in this study. The measurement results of ACSM or AMS including the concentrations of sulfate, nitrate, ammonium salt and organics. These results were shown in Fig.3 in the manuscript.

*NH3 was estimated in the chamber according to Table 2. Why NH3 was not measured experimentally vs time? This is very important (same for SO2). I suggest to use "experiment without NH3 added to the chamber" instead of "NH3 poor"? It's confusing and make it difficult for comparison with other data?*

**Response:** As we mentioned earlier, the concentrations on $NH_3$ were not measured due to lack of analytical instruments. Besides, as we mentioned in the manuscript, there was $NH_3$ present in the background air in the chamber derived from the partitioning of the deposited ammonium sulfate and nitrate on the chamber wall when humid air was introduced. Based on the results of experiment STN, the background $NH_3$ was estimated to be around 8 ppb. With this in mind, the experiments carried out without $NH_3$ added were considered "$NH_3$-poor" experiments in this study, while experiments with $NH_3$ added were considered "$NH_3$-rich" experiments.

*Line 314. Should be Table 2.*

**Response:** Thanks for the reminding! Corresponding revision has been made in the revised manuscript.

**Revision in the manuscript:**

**Fig. 4 caption:** "Table 1"

**To:** "Table 2"

**References**

Anttila, T., Kiendler-Scharr, A., Tillmann, R., and Mentel, T. F.: On the reactive uptake of gaseous compounds by organic-coated aqueous aerosols: Theoretical analysis and application to the heterogeneous hydrolysis of $N_2O_5$, J. Phys. Chem. A, 110, 10435-10443, doi: 10.1021/jp062403c, 2006.

Atkinson, R., Baulch, D. L., Cox, R. A., Crowley, J. N., Hampson, R. F., Hynes, R. G., Jenkin, M. E., Rossi, M. J., and Troe, J.: Evaluated kinetic and photochemical data for atmospheric chemistry: Volume I - gas phase reactions of Ox, HOx, NOx and SOx species, Atmos. Chem. Phys., 4, 1461-1738, doi: doi:10.5194/acp-4-1461-2004, 2004.

Bauduin, S., Clarisse, L., Hadji-Lazaro, J., Theys, N., Clerbaux, C., and Coheur, P. F.: Retrieval of near-surface sulfur dioxide ($SO_2$) concentrations at a global scale using IASI satellite observations, Atmos. Meas. Tech., 9, 721-740,

doi:10.5194/amt-9-721-2016, 2016.

Carslaw, K. S., Clegg, S. L., and Brimblecombe, P.: A thermodynamic model of the system $HCl$-$HNO_3$-$H_2SO_4$-$H_2O$, including solubilities of HBr, from less-than-200 to 328 K, J. Phys. Chem., 99, 11557-11574, doi:10.1021/j100029a039, 1995.

Chu, B., Hao, J., Takekawa, H., Li, J., Wang, K., and Jiang, J.: The remarkable effect of $FeSO_4$ seed aerosols on secondary organic aerosol formation from photooxidation of α-pinene/NOx and toluene/NOx, Atmos. Environ., 55, 26-34, doi:10.1016/j.atmosenv.2012.03.006, 2012.

Chu, B., Liu, Y., Li, J., Takekawa, H., Liggio, J., Li, S.-M., Jiang, J., Hao, J., and He, H.: Decreasing effect and mechanism of $FeSO_4$ seed particles on secondary organic aerosol in α-pinene photooxidation, Environ. Pollut., 193, 88-93, doi:10.1016/j.envpol.2014.06.018, 2014.

Clegg, S. L., Brimblecombe, P., and Wexler, A. S.: Thermodynamic model of the system $H^+$-$NH_4^+$-$SO_4^{2-}$-$NO_3^-$-$H_2O$ at tropospheric temperatures, J. Phys. Chem. A, 102, 2137-2154, doi:10.1021/jp973042r, 1998.

Clegg, S. L., and Brimblecombe, P.: Comment on the "Thermodynamic dissociation constant of the bisulfate ion from Raman and ion interaction modeling studies of aqueous sulfuric acid at low temperatures", J. Phys. Chem. A, 109, 2703-2706, doi:10.1021/jp0401170, 2005.

Dong, W. X., Xing, J., and Wang, S. X.: Temporal and spatial distribution of anthropogenic ammonia emissions in China: 1994–2006, Huanjingkexue, 31, 1457-1463, doi: 10.13227/j.hjkx.2010.07.008, 2010.

Hallquist, M., Stewart, D. J., Stephenson, S. K., and Cox, R. A.: Hydrolysis of N2O5 on sub-micron sulfate aerosols, Phys. Chem. Chem. Phys., 5, 3453-3463, doi: 10.1039/b301827j, 2003.

Hildebrandt Ruiz, L., Paciga, A. L., Cerully, K. M., Nenes, A., Donahue, N. M., and Pandis, S. N.: Formation and aging of secondary organic aerosol from toluene: changes in chemical composition, volatility, and hygroscopicity, Atmos. Chem. Phys., 15, 8301-8313, doi:10.5194/acp-15-8301-2015, 2015.

Hu, J. H., and Abbatt, J. P. D.: Reaction probabilities for $N_2O_5$ hydrolysis on sulfuric acid and ammonium sulfate aerosols at room temperature, J. Phys. Chem. A, 101, 871-878, doi: 10.1021/jp9627436, 1997.

Li, C., Marufu, L. T., Dickerson, R. R., Li, Z., Wen, T., Wang, Y., Wang, P., Chen, H., and Stehr, J. W.: In situ measurements of trace gases and aerosol optical properties at a rural site in northern China during East Asian Study of Tropospheric Aerosols: An International Regional Experiment 2005, J. Geophys. Res.- Atmos., 112, D22S04, doi:10.1029/2006JD007592, 2007.

Li, W. J., Shao, L. Y., Shi, Z. B., Chen, J. M., Yang, L. X., Yuan, Q., Yan, C., Zhang, X. Y., Wang, Y. Q., Sun, J. Y., Zhang, Y. M., Shen, X. J., Wang, Z. F., and Wang, W. X.: Mixing state and hygroscopicity of dust and haze particles before leaving Asian continent, J. Geophys. Res.- Atmos., 119, 1044-1059, doi:10.1002/2013jd021003, 2014.

Lu, Z., Streets, D. G., Zhang, Q., Wang, S., Carmichael, G. R., Cheng, Y. F., Wei, C., Chin, M., Diehl, T., and Tan, Q.: Sulfur dioxide emissions in China and sulfur trends

in East Asia since 2000, Atmos. Chem. Phys., doi:10, 6311-6331, 10.5194/acp-10-6311-2010, 2010.

Meyer, N. K., Duplissy, J., Gysel, M., Metzger, A., Dommen, J., Weingartner, E., Alfarra, M. R., Prevot, A. S. H., Fletcher, C., Good, N., McFiggans, G., Jonsson, A. M., Hallquist, M., Baltensperger, U., and Ristovski, Z. D.: Analysis of the hygroscopic and volatile properties of ammonium sulphate seeded and unseeded SOA particles, Atmos. Chem. Phys., 9, 721-732, doi: 10.5194/acp-9-721-2009, 2009.

Odum, J. R., Jungkamp, T. P. W., Griffin, R. J., Forstner, H. J. L., Flagan, R. C., and Seinfeld, J. H.: Aromatics, reformulated gasoline, and atmospheric organic aerosol formation, Environ. Sci. & Technol., 31, 1890-1897, doi:10.1021/es960535l, 1997.

Sato, K., Hatakeyama, S., and Imamura, T.: Secondary organic aerosol formation during the photooxidation of toluene: NOx dependence of chemical composition, J. Phys. Chem. A, 111, 9796-9808, doi:10.1021/jp071419f, 2007.

Sun, Y., Wang, Y. S., and Zhang, C. C.: Measurement of the vertical profile of atmospheric $SO_2$ during the heating period in Beijing on days of high air pollution, Atmos. Environ., 43, 468-472, doi:10.1016/j.atmosenv.2008.09.057, 2009.

Takekawa, H., Minoura, H., and Yamazaki, S.: Temperature dependence of secondary organic aerosol formation by photo-oxidation of hydrocarbons, Atmos. Environ., 37, 3413-3424, doi:10.1016/s1352-2310(03)00359-5, 2003.

Wang, L., Wen, L., Xu, C., Chen, J., Wang, X., Yang, L., Wang, W., Yang, X., Sui, X., Yao, L., and Zhang, Q.: HONO and its potential source particulate nitrite at an urban site in North China during the cold season, Sci. Total Environ., 538, 93-101, doi:10.1016/j.scitotenv.2015.08.032, 2015a.

Wang, S. W., Zhang, Q., Martin, R. V., Philip, S., Liu, F., Li, M., Jiang, X. J., and He, K. B.: Satellite measurements oversee China's sulfur dioxide emission reductions from coal-fired power plants, Environ. Res. Lett., 10, 9, doi:10.1088/1748-9326/10/11/114015, 2015b.

Zhang, Q., Shen, Z. X., Cao, J. J., Zhang, R. J., Zhang, L. M., Huang, R. J., Zheng, C. J., Wang, L. Q., Liu, S. X., Xu, H. M., Zheng, C. L., and Liu, P. P.: Variations in $PM_{2.5}$, TSP, BC, and trace gases ($NO_2$, $SO_2$, and $O_3$) between haze and non-haze episodes in winter over Xi'an, China, Atmos. Environ., 112, 64-71, doi:10.1016/j.atmosenv.2015.04.033, 2015.